# Resampling and ensemble techniques for improving ANN-based high flow forecast accuracy

Everett Snieder[1], Karen Abogadil[1], and Usman T. Khan[1]

[1]Department of Civil Engineering, York University, 4700 Keele St, Toronto ON, Canada, M3J 1P3

**Correspondence:** Usman T. Khan (usman.khan@lassonde.yorku.ca)

**Abstract.** Data-driven flow forecasting models, such as Artificial Neural Networks (ANNs), are increasingly featured in research for their potential use in operational riverine flood warning systems. However, the distributions of observed flow data are imbalanced, resulting in poor prediction accuracy on high flows, both in terms of amplitude and timing error. Resampling and ensemble techniques have shown to improve model performance on imbalanced datasets. However, the efficacy of these methods (individually or combined) has not been explicitly evaluated for improving high flow forecasts. In this research, we systematically evaluate and compare three resampling methods: random undersampling (RUS), random oversampling (ROS), and synthetic minority oversampling technique for regression (SMOTER); and four ensemble techniques: randomised weights and biases, Bagging, adaptive boosting (AdaBoost), least squares boosting (LSBoost); on their ability to improve high stage prediction accuracy using ANNs. These methods are implemented both independently and in combined, hybrid techniques, where the resampling methods are embedded within the ensemble methods. This systematic approach for embedding resampling methods are novel contributions. This research presents the first analysis of the effects of combining these methods on high stage prediction accuracy. Data from two Canadian watersheds (the Bow River in Alberta, and the Don River in Ontario), representing distinct hydrological systems, are used as the basis for the comparison of the methods. The models are evaluated on overall performance, and on typical and high stage subsets. The results of this research indicate that resampling produces marginal improvements to high stage prediction accuracy, whereas ensemble methods produce more substantial improvements, with or without resampling. Many of the techniques used produced an asymmetric trade-off between typical and high stage performance; reduction of high stage error resulted in disproportionately larger error on typical stage. The methods proposed in this study highlight the diversity-in-learning concept and help support for future studies on adapting ensemble algorithms for resampling. This research contains many of the first instances of such methods for flow forecasting and moreover, their efficacy to address the imbalance problem and heteroscedasticity, which are commonly observed in high flow and flood forecasting models.

# 1 Introduction

Data-driven models such as artificial neural networks (ANNs) have been widely and successfully used over the last three decades for hydrological forecasting applications (Govindaraju, 2000; Abrahart et al., 2012; Dawson and Wilby, 2001). However, some studies have noted that these models can exhibit poor performance during high flow (or stage) hydrological events (Sudheer et al., 2003; Abrahart et al., 2007; de Vos and Rientjes, 2009), with poor performance manifesting as late predictions (i.e., timing error), under-predictions, or both. For flow forecasting applications such as riverine flood warning systems, the accuracy of high stage predictions are more important than that of typical stage. One cause of poor model accuracy on high stage is the scarcity of representative sample observations available with which to train such models (Moniz et al., 2017a). This is because stage data typically exhibits a strong positive skew, referred to as an imbalanced domain; thus, there may only be a small number of flood observations within decades of samples. Consequently, objective functions that are traditionally used for training ANNs (e.g., mean squared error (MSE), sum of squared error (SSE), etc.), that equally consider all samples, are biased towards values that occur most frequently and reflected by poor model performance on high flow or stage observations (Pisa et al., 2019). Sudheer et al. (2003) also point out that such objective functions are not optimal for non-normally distributed data. This problem is exacerbated when such metrics are also used to assess model performance; regrettably, such metrics are the most widely used in water resources applications (Maier et al., 2010). As a result, studies that assess models using traditional performance metrics risk overlooking deficiencies in high stage performance.

Real-time data-driven flow forecasting models frequently use antecedent input variables (also referred to as autoregressive inputs) for predictions. Several studies have attributed poor model prediction on high stage to model over-reliance on antecedent variables (Snieder et al., 2020; Abrahart et al., 2007; de Vos and Rientjes, 2009; Tongal and Booij, 2018). Consequently, the model predictions are similar to the most recent antecedent conditions, sometimes described as a lagged prediction (Tongal and Booij, 2018). In other words, the real-time observed stage at the target gauge is used as the predicted value for a given lead time. This issue is closely linked to the imbalanced domain problem as frequently occurring stage values typically exhibit low temporal variability compared to infrequent, high stage values; this phenomenon is further described in Sect. 2.

Improving the accuracy of high stage or flow forecasts has been the focus of many studies. Several studies have examined the use of preprocessing techniques to improve model performance. Sudheer et al. (2003) propose using a Wilson-Hilferty transformation to change the skewed distribution of stage data. The study found that transforming the target data reduces annual peak flow error produced by ANN-based daily flow forecasting models. Wang et al. (2006) evaluate three strategies for categorising streamflow samples, based on a fixed value flow threshold, unsupervised clustering, and periodicity; separate ANN models are trained to predict each flow category and combined to form a final prediction. The periodicity-based ANN, which detects periodicity from the autocorrelation function of the target variable, is found to perform the best out of the three schemes considered. Fleming et al. (2015) address the issue of poor high flow performance by isolating a subset of daily high flows by thresholding based on a fixed value. By doing so, traditional objective functions (e.g., MSE) become less influenced by the imbalance of the training dataset. ANN-based ensembles trained on high flows are found to perform well, though

the improvements to high flow accuracy are not directly quantified, as the high flow ensemble is not compared directly to a counterpart trained using the full training dataset.

An alternative approach to improving high flow forecast accuracy has been to characterise model error as having amplitude and temporal components (Seibert et al., 2016). Abrahart et al. (2007) use a specialised learning technique in which models are optimised based on a combination of root mean square error (RMSE) and a timing error correction factor, which is found to improve model timing for short lead-times, but have little impact on higher lead times. de Vos and Rientjes (2009) use a similar approach, in which models that exhibit a timing error are penalised during calibration. The technique is found to generally reduce timing error at the expense of amplitude error.

Finally, there is considerable evidence that ensemble-based and resampling techniques to improve prediction accuracy of infrequent samples (Galar et al., 2012). Ensemble methods, such as bootstrap aggregating (Bagging) and boosting, are known for their ability to improve model generalisation. Such methods are widely used in classification studies and are increasingly being adapted for regression tasks (Moniz et al., 2017b). However, ensemble methods alone do not directly address the imbalance problem, as they typically do not explicitly consider the distribution of the target dataset. Thus, ensemble methods are often combined with preprocessing strategies to address the imbalance problem (Galar et al., 2012). Resampling, which is typically used as a preprocessing method, can be used to create more uniformly distributed target dataset or generate synthetic data with which to train models (Moniz et al., 2017a). Resampling also promotes diversity-in-learning when embedded in ensemble algorithms (rather than used as a preprocessing strategy). Examples of such combinations appear in machine learning literature, but are typically developed for ad hoc applications (Galar et al., 2012).

However, the efficacy of these methods (a combination of resampling strategies with ensemble methods) has not been systematically investigated for flow forecasting applications. While previous studies have provided comparisons of ensemble methods, none have explicitly studied their effects on high flow prediction accuracy, which has only received little attention within the context of the imbalance problem in general. Additionally, previous research uses resampling as a preprocessing technique, whereas in this research, resampling is embedded within the ensembles to promote diversity-in-learning. Thus, the main objective of this research is to develop a systematised framework for combining several different resampling and ensemble techniques with the aim to improve high flow forecasts using ANNs. Three resampling techniques: random undersampling (RUS), random oversampling (ROS), and synthetic minority oversampling technique for regression (SMOTER) and four ensemble algorithms: randomised weights and biases (RWB), Bagging, adaptive boosting for regression (AdaBoost), and least-squares boosting (LSBoost) will be investigated to address the issues related to high flow forecasts, i.e., the imbalanced domain problem and heteroscedasticity. Each combination of these methods will be explicitly evaluated on their ability to improve model performance on high stage (infrequent) data subsets along with the typical (frequent) data subsets. Such a framework and comparison, to address the imbalanced domain, has not been presented in existing literature. Lastly, while only selected resampling and ensemble techniques are presented, many of which are the first instances of their use for high flow forecasting, this proposed framework may easily be expanded to resampling and ensemble strategies beyond those included in this research.

The remainder of the manuscript is organised as follows: first, in in Sect. 2 we present the baseline ANN flow forecast models, which are used as the individual learners for the ensembles, for two Canadian watersheds, followed by a performance analysis of these models to highlight the imbalance domain problem and illustrates the heteroscedasticity of baseline model residuals. The two watersheds, with differing hydrological characteristics, but both prone to riverine floods, are the Bow River watershed (in Alberta), and the Don River watershed (in Ontario). Sect. 3 provides a review and applications of each resampling method and ensemble technique, followed by a description of the implementation of each approach in this research, and model evaluation methods. Lastly, Sect. 4 includes the results and discussion from the two case studies.

## 2 Early investigations

The following section provides descriptions for the two watersheds under study. The parametrisation of the single ANN models to predict stage in each watershed (referred to as the individual learners) is described. The output of the individual learners are used to exemplify the inability of these ANNs to accurately predict high stage (from both an amplitude and temporal error perspective) and to illustrate the imbalance problem.

### 2.1 Study area

The Bow and Don Rivers are featured as case studies in this research to evaluate methods for improving the accuracy of high stage data-driven forecasts. The Bow River, illustrated in Fig. 1 (a), begins in the Canadian Rockies mountain range and flows eastward through the City of Calgary, where it is joined by the Elbow River. The Bow River's flow regime is dominated by glacial and snowmelt processes which produce annual seasonality. The Bow River watershed has an area of approximately $7,700 km^2$ upstream of the target stage monitoring station in Calgary and consists of predominantly natural and agricultural land cover. The City of Calgary has experienced several major floods (recently in 2005 and 2013) and improvements to flow forecasting models have been identified as a key strategy for mitigating flood damage Khan et al. (2018).

The Don River, illustrated in Fig. 1 (b), begins in the Oak Ridges Moraine and winds through the Greater Toronto Area until it meets Lake Ontario in downtown Toronto. The $360 km^2$ Don River watershed is heavily urbanised which results in the high stage seen in the River to be attributable to the direct runoff following intense rainfall events. Its urbanised landscape has also contributed to periodic historical flooding (Toronto and Region Conservation Authority, 2020a). Persistent severe flooding (recently in 2005 and 2013) have motivated calls for further mitigation strategies such as improved flow forecast models and early warning systems (Nirupama et al., 2014).

Data from November to April and November to December were removed from the Bow and Don River datasets, prior to any analysis; these periods are associated with ice conditions. The histograms in Figure 2 illustrate the imbalanced domains of the target stage for both rivers. A high stage threshold ($\Theta_{HS}$) is defined, which is used to distinguish between typical and high stage. Stage values greater than the threshold are referred to as high stage ($q_{HS}$) while stage below the threshold, as typical stage ($q_{TS}$). Target stage statistics for the Bow and Don Rivers are provided for the complete stage distribution, as well as the $q_{TS}$ and $q_{HS}$ subsets, in Table 1.

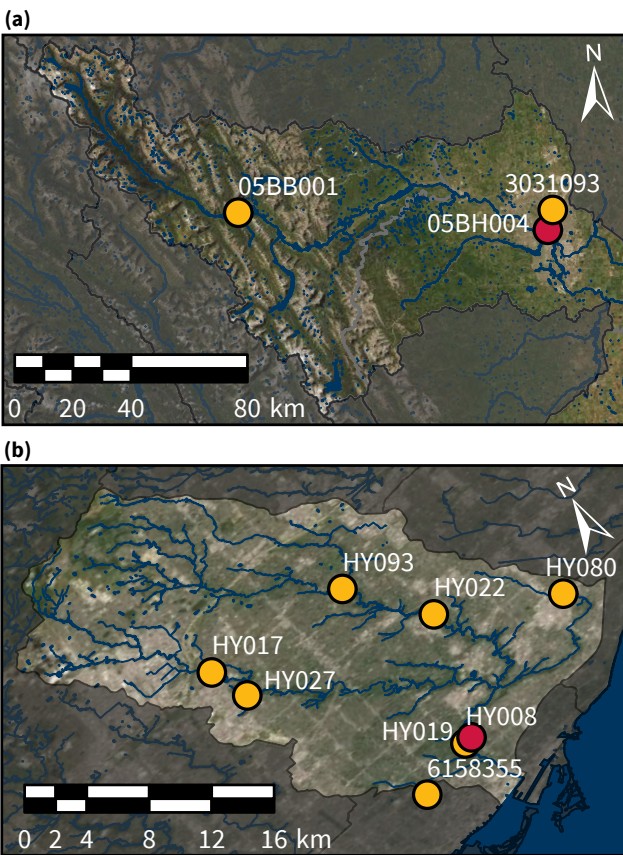

**Figure 1.** Bow (a) and Don (b) River basins upstream of Calgary and Toronto, respectively. Surface watercourses and waterbodies are shown in blue. The target stage monitoring stations are red while upstream hydrometeorological monitoring stations (stage, precipitation, and temperature) are yellow. Aerial imagery obtained from © Esri (Esri, 2020). Surface water and watershed boundaries obtained from © Scholars GeoPortal (DMTI Spatial Inc., 2014a, b, c, 2019) and the © TRCA (Toronto and Region Conservation Authority, 2020b)

**Table 1.** Target variable statistics for the Bow and Don River watersheds.

| River | Subset | Mean | Min. | Max. | Skew. | Var. |
| --- | --- | --- | --- | --- | --- | --- |
| | | [m] | [m] | [m] | [-] | [m$^2$] |
| Bow | $q$ | 1.28 | 0.92 | 3.07 | 1.18 | 0.067 |
| | $q_{TS}$ | 1.18 | 0.92 | 1.47 | 0.21 | 0.022 |
| | $q_{HS}$ | 1.69 | 1.47 | 3.07 | 1.85 | 0.039 |
| Don | $q$ | 77.62 | 77.51 | 79.21 | 3.78 | 0.018 |
| | $q_{TS}$ | 77.58 | 77.51 | 77.67 | 0.59 | 0.0017 |
| | $q_{HS}$ | 77.82 | 77.68 | 79.21 | 2.99 | 0.034 |

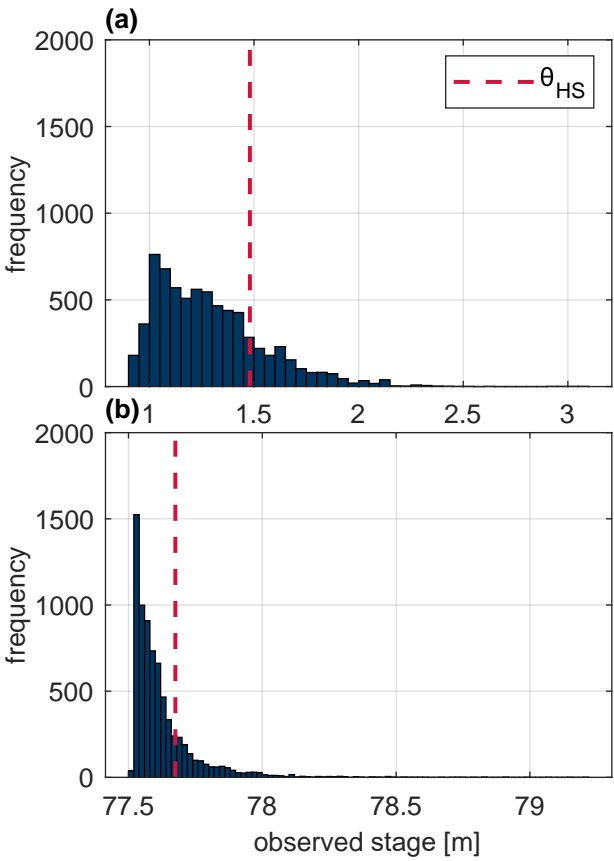

**Figure 2.** Histograms of observed stage for the (a) Bow River 6-hour stage and (b) Don River hourly stage. The dashed red line indicates the fixed threshold used to distinguish between typical and high stage values.

The use of a fixed threshold for distinguishing between common (frequent) and rare (infrequent) samples is used both in flow forecasting (Crochemore et al., 2015; Razali et al., 2020; Fleming et al., 2015) and in more general machine learning studies that are focused on the imbalance problem (Moniz et al., 2017a). In this research, the high stage threshold is simply and arbitrarily taken as the 80th percentile value of the observed stage. The threshold value is ideally derived from the physical characteristics of the river (i.e., the stage at which water exceeds the bank or associated with a specified return period); unfortunately this site-specific information is not readily available for the subject watersheds. An important consideration to make while selecting a $\Theta_{HS}$ value is that it produces a sufficient number of high stage samples; too few samples risks overfitting and poor generalisation. The distinction between typical and high stage is used in some of the resampling techniques in Sect. 3.1 and for assessing model performance in Sect. 3.4.

**Table 2.** Individual learner ANN model description used for both watersheds.

| | |
|---|---|
| **Model class** | Artificial neural network |
| **Architecture** | Multi-layer perceptron |
| **IVS** | Partial correlation |
| **Hidden neurons** | 10 |
| **Activation function** | Tanh (hidden layer), Linear (output layer) |
| **Training algorithm** | Levenburg-Marquardt backpropagation |
| **Stopping criteria** | Validation dataset |

**Table 3.** Input variables for the Bow and Don Rivers.

| Catchment | Variable | Station ID | Statistics | Data source | Lag times |
|---|---|---|---|---|---|
| **Bow River** | Water level | 05BB001, 05BH004* | Max, min, mean 6-hour | Water Survey of Canada | 0:11 |
| **6-hour timestep** | Precipitation | 031093 | Cumulative 6-hour | City of Calgary | 0:11 |
| **24-hour forecast** | Temperature | 031093 | Max, min, mean 6-hour | City of Calgary | 0:11 |
| **Don River** | Water level | HY017, HY019*, HY022, HY080, HY093 | Hourly | TRCA | 0:5 |
| **1-hour timestep** | Precipitation | HY008, HY927 | Hourly | TRCA | 0:11 |
| **4-hour forecast** | Temperature | 6158355 | Hourly | Environment Canada | 0:5 |

\* indicates target station

## 2.2 Individual learner description

The individual learner (sometimes called the base model, or base learner) for both systems use upstream hydro-meteorological inputs (stage, precipitation, and temperature) to predict the downstream stage (the target variable). The multi-layer perception (MLP) ANN is used as the individual learner for this study and the selected model hyperparameters are summarised in Table 2. The MLP-ANN was chosen as the individual learner because it is the most commonly used machine learning architecture for predicting water resources variables in river systems (Maier et al., 2010). The individual learner can be used for discrete value prediction or as a member of an ensemble, in which a collection of models are trained and combined to generate predictions. Each ANN has a hidden layer of 10 neurons; a grid-search of different hidden layer sizes indicated that larger numbers of hidden neurons have little impact on the ANN performance. Thus, to prevent needlessly increasing model complexity, a small hidden layer is favoured. The number of training epochs is determined using early-stopping (also called stop-training), which is performed by dividing the calibration data into training and validation subsets; training data is used to tune the ANN weights and biases whereas the validation performance is used to determine when to stop training (Anctil and Lauzon, 2004). For this study, the optimum number of epochs is assumed if the error on the validation set increases for 5 consecutive epochs. Early-stopping is a common technique for achieving generalisation and preventing overfitting (Anctil and Lauzon, 2004). Of the available data for each watershed, 60% is used for training, 20% for validation, and 20% for testing (the independent dataset). K-fold cross-validation (KFCV) is used to evaluate different continuous partitions of training and testing data, and is explained

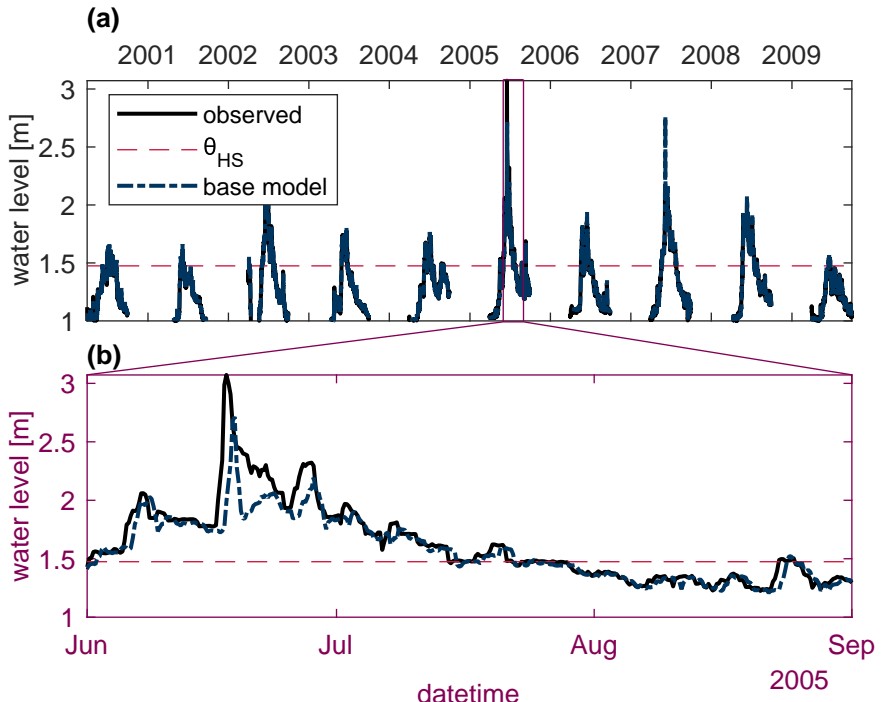

**Figure 3.** Observed and individual learner stage predictions for the Bow River system for all 10 years of available stage (a) and a 3 month subset which contains particularly high stage (b), to better distinguish between the two hydrographs. The dashed red line indicates the fixed threshold used to distinguish between typical and high stage values.

in greater detail in Sect. 3.4.2. The Levenberg–Marquardt algorithm was used to train the individual learners, because of its speed of convergence and reliability (Lauzon et al., 2006; Maier and Dandy, 2000; Tongal and Booij, 2018). The full set of input and target variables used for both catchments are summarised in Table 3. For both rivers, the input variables are used to

forecast the target variable 4 timesteps in advance, i.e., for the Bow River, the model forecasts 24 hours in the future, whereas for the Don River, the model forecasts 4 hours in the future. Some of the input variables used in the Bow River model, including the minimum, mean, and maximum statistics, are calculated by coarsening hourly data to a 6-hour timestep. Several lagged copies of each input variable are used, which is common practice for ANN-based hydrological forecasting models (Snieder et al., 2020; Abbot and Marohasy, 2014; Fernando et al., 2009; Banjac et al., 2015). For example, to forecast $x_t$ by 4 timesteps,

$x_{t-4}$, $x_{t-5}$, $x_{t-6}$, etc. may be used as an input variables, as these variables are recorded automatically, in real-time.

The Partial Correlation (PC) input variable selection (IVS) algorithm is used to to determine the most suitable inputs for each model from the larger candidate set (He et al., 2011; Sharma, 2000). Previous research for the Don and Bow Rivers found that PC is generally capable of removing non-useful inputs in both systems, achieving reduced computational demand and improved model performance (Snieder et al., 2020). The simplicity and computational efficiency of the PC algorithm method

makes it an appealing IVS algorithm for this application. The 25 most useful inputs amongst all the candidates listed in Table

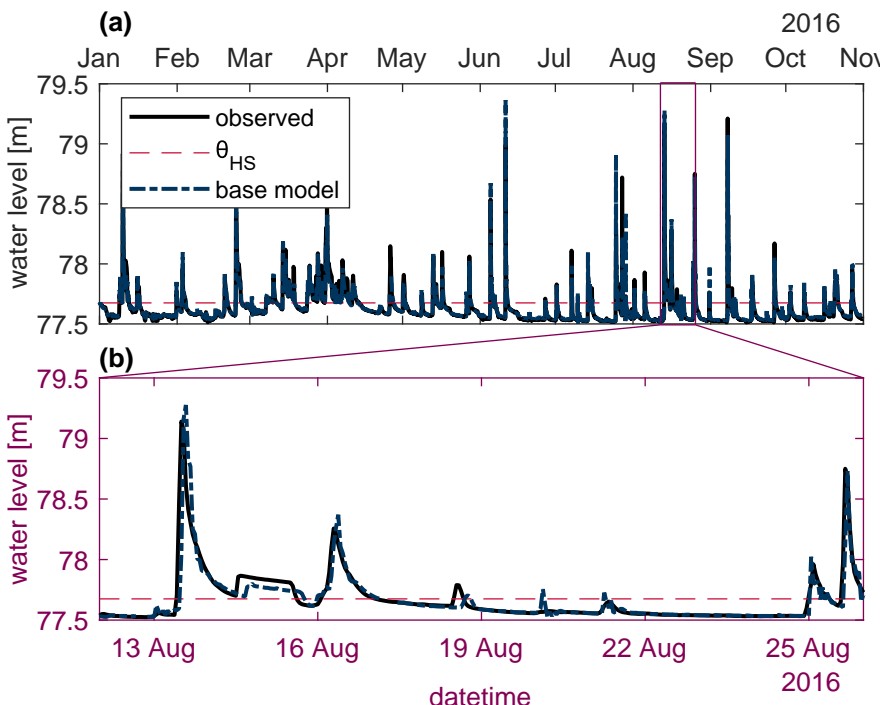

**Figure 4.** Observed and individual learner stage predictions for the Don River system for all 10 months of available stage (a) and a 14 day subset which contains particularly high stage (b), to better distinguish the two hydrographs. The dashed red line indicates the fixed threshold used to distinguish between typical and high stage values.

3, determined by the PC algorithm, are used in the models for each watershed. A complete list of selected inputs is shown in Appendix A.

The Bow and Don River individual learners produce coefficients of Nash-Sutcliffe efficiency (CE) greater than 0.95 and 0.75, respectively. These scores are widely considered by hydrologists to indicate good performance (Crochemore et al., 2015).
However, closer investigation of the model performance reveals that high stage samples consistently exhibit considerable error. Such is plainly visible when comparing the observed hydrographs with the individual learner predictions, as shown in Figs. 3 and 4, for the Bow and Don Rivers, respectively. Plotting the individual learner residuals against the observed stage, as in Fig. 5 (a and b) illustrates how the variance of the residuals about the expected mean of 0 increases with the increasing stage magnitude; Fleming et al. (2015) also describe the heteroscedastic nature of flow prediction models. This region of high stage
also exhibits amplitude errors in the excess of 1 meter, casting doubt on the suitability of these models for flood forecasting applications. In Fig. 5 (b and c) the normalised inverse frequency of each sample point is plotted against the stage gradient, illustrating how the most frequent stage values typically have a low gradient with respect to the forecast lead time, given by $(q_{t+L} - q_t)/L$. Note that the inverse frequency is determined using 100 histogram bins. Thus, when such a relationship exists, it is unsurprising that model output predictions are similar to the most recent autoregressive input variable. Previous work that

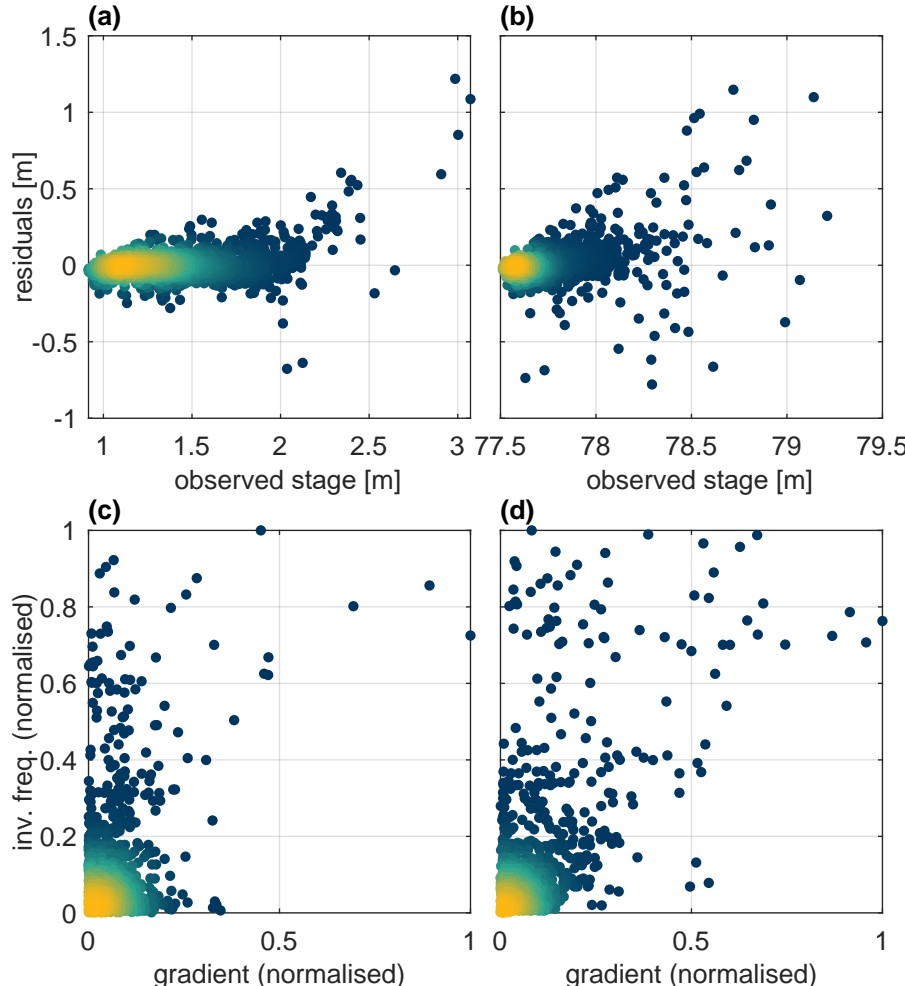

**Figure 5.** Baseline model residuals versus observed stage for the Bow (a) and Don (b) River systems. Inverse frequency versus gradient across 4 time steps for the Bow (c) and Don (d) River target variables. Colouring indicates normalised scatter point density.

analysed trained ANN models for both subject watersheds demonstrates how the most recent autoregressive input variable is the most important variable for accurate stage predictions (Snieder et al., 2020).

     Without accounting for the imbalanced nature of stage data, data-driven models are prone to inadequate performance similar to that of the individual learners described above. Consequently, such models may not be suitable for flood related applications such as early flood warning systems. The following section describes, and reviews resampling and ensemble methods, which

are proposed as solutions to the imbalance problem, which manifests as poor performance on high stage samples, relative to typical stage.

## 3  Review and description of methods for handling imbalanced target datasets

Many strategies have been proposed for handling imbalanced domains, which can be broadly categorised into three approaches: specialised preprocessing, learning methods, and combined methods (Haixiang et al., 2017; Moniz et al., 2018). According to a comprehensive review of imbalanced learning strategies resampling and ensemble methods are among the most popular techniques employed (Haixiang et al., 2017). Specifically, a review of 527 papers on imbalanced classification found that a resampling technique was used 156 times (Haixiang et al., 2017). From the same review, 218 of the 527 papers used an ensemble technique such as Bagging or boosting. Many of the studies reviewed used combinations of available techniques and often propose novel hybrid approaches that incorporate elements from several algorithms. Since it is impractical to compare every unique algorithm that has been developed for handling imbalanced data, the scope of this research adheres to relatively basic techniques and combinations of resampling and ensemble methods. The following sections describe the resampling and ensemble methods used in this research. The review attempts to adhere to hydrological studies that feature each of the methods, however, when this is not always possible, examples from other fields are presented.

First, it is important to distinguish between the data imbalance addressed in this study and cost-sensitive imbalance. Imbalance in datasets can be characterised as a combination of two factors: imbalanced distributions of samples across the target domain and imbalanced user interest across the domain. Target domain imbalance is related solely to the native distribution of samples while cost-sensitivity occurs when costs vary across the target domain. While both types of imbalance are relevant to the flow forecasting application of this research, cost-sensitive methods are complex and typically involve developing a relationship between misprediction and tangible costs, for example, property damage (Toth, 2016). Cost-sensitive learning is outside the scope of this research, which is focused on reducing high stage errors due to the imbalanced nature of the target stage data.

### 3.1  Resampling techniques

Resampling is widely used in machine learning to create subsets of the total available data with which to train models. Resampling is typically used as a data preprocessing technique (Brown et al., 2005; Moniz et al., 2017a). However, in our research, resampling is embedded in the ensemble algorithms, as to promote diversity amongst the individual learners. This following section discusses examples of resampling, whether used for preprocessing or used within the learning algorithm. Pseudocode for each resampling method is provided in Appendix B.

#### 3.1.1  Random undersampling

RUS is performed by subsampling a number of frequent cases equal to the number of infrequent cases, such that there are an even amount in each category and achieve a more balanced distribution compared to the original set. As a result, all of the rare cases are used for training, while only a fraction of the normal cases are used. RUS is intuitive for classification problems; for two-class classification, the majority class is undersampled such that the number of samples drawn from each class is equal to the number of samples in the minority class (Yap et al., 2014). However, RUS is less straightforward for regression, as it

requires continuous data first to be categorised, as to allow for an even number of samples to be drawn from each category. Categories must be selected appropriately such that they are continuous across the target domain and each category contains a sufficient number of samples to allow for diversity in the resampled dataset (Galar et al., 2013). Undersampling is scarcely used in hydrological forecasting applications, despite seeing widespread use in classification studies. Ruhana et al. (2014) demonstrate an application of fuzzy-based RUS for categorical flood risk support vector machine (SVM) based classification, which is motivated by the imbalanced nature of flood data. RUS is found to outperform both ROS and synthetic minority oversampling technique (SMOTE) on average across 5 locations.

In this research, $N$ available stage samples are categorised into $N_{TS}$ typical stage and $N_{HS}$ high stage based on the threshold $\Theta_{HS}$. The undersampling scheme draws $N_{HS}$ with replacement from each of the subsets, such that there are an equal number of each category. RUS can be performed with or without replacement; the former provides greater diversity when resampling is repeated several times, and thus this approach is selected for the present research.

### 3.1.2  Random oversampling

ROS simply consists of oversampling rare samples, thus modifying the training sample distribution through duplication (Yap et al., 2014). ROS is procedurally similar to RUS, also aiming to achieve a common number of frequent and infrequent samples. Instead of subsampling the typical stage, high stage values are resampled with replacement so that the number of samples matches that of the typical stage set. The duplication of high stage samples in the training dataset increases their relative contribution to the model's objective function during calibration. Compared to undersampling, oversampling is advantaged such that more samples in the majority class are utilised. The drawbacks of this approach are that there is an increased computational cost. There are few examples of ROS applications in water resources literature; studies tend to favour SMOTE, which is discussed in the following section. Saffarpour et al. (2015) use oversampling to address the class imbalance of binary flood data; surprisingly, oversampling was found to decrease classification accuracy compared to the raw training dataset. Recently, Zhaowei et al. (2020) applied oversampling for vehicle traffic flow, as a response to the imbalance of the training data.

For ROS, as with RUS, $N$ available stage samples are categorised into $N_{TS}$ typical stage and $N_{HS}$ high stage samples based on the threshold $\Theta_{HS}$. The oversampling scheme draws $N_{TS}$ with replacement from each of the subsets, such that there are an equal number of each category. ROS is distinguished from RUS in that it produces a larger sample set that inevitably contains duplicated high stage values.

### 3.1.3  Synthetic minority oversampling technique for regression

SMOTER is a variation of the SMOTE classification resampling technique introduced by (Chawla et al., 2002) that bypasses excessive duplication of samples by generating synthetic samples, which unlike duplication, creates diversity within the ensembles. SMOTE is widely considered as an improvement over simple ROS as the increased diversity help prevents overfitting (Ruhana et al., 2014). For a given sample, SMOTE generates synthetic samples by randomly selecting one of k nearest points, determined using k-nearest neighbours (KNN), and sampling a value at a linear distance between the two neighbouring points. The original SMOTE algorithm was developed for classification tasks; Torgo et al. (2013) developed the SMOTER variation,

which is an adaptation of SMOTE for regression. SMOTER uses a fixed threshold to distinguish between 'rare' and 'normal' points. In addition to oversampling synthetic data, SMOTER also randomly undersamples normal values, to achieve the desired ratio between rare and normal samples. The use of SMOTE in the development of models that predict river stage is only being recently attempted. Atieh et al. (2017) use two methods for generalisation: Dropout and SMOTER; these were applied to ANN models that predicted the flow duration curves for ungauged basins. They found that SMOTER reduced the number of outlier predictions, whereas both approaches resulted in the improved performance of the ANN models. Wu et al. (2020) used SMOTE resampling in combination with AdaBoosted sparse Bayesian models. The combination of these methods resulted in improved model accuracy compared to previous studies using the same dataset. Razali et al. (2020) used SMOTE with various Bayesian network and machine learning techniques, including decision trees, KNN and SVM. Each technique is applied to an imbalanced classified flood dataset (flood flow and non-flood flow categories); the SMOTE decision tree model achieved the highest classification accuracy. SMOTE decision trees have also been applied for estimating the pollutant removal efficiency of bioretention cells. Wang et al. (2019a) found that decision trees developed with SMOTE had the highest accuracy for predicting pollutant removal rates; the authors attribute the success of SMOTE to its ability to prevent the majority class from dominating the fitting process. Sufi Karimi et al. (2019) employ SMOTER resampling for stormwater flow prediction models. Their motivation for resampling is flow dataset imbalance and data sparsity. Several configurations are considered with varying degrees of oversampled synthetic and undersampled data. The findings of the study indicate that increasing the oversampling rate tends to improve model performance compared to the non-resampled model, while increasing the undersampling rate produces a marginal improvement. Collectively, these applications of SMOTE affirm its suitability for mitigating the imbalance problem in the flood forecasting models featured in this research.

SMOTER is adapted in this research following the method described by (Torgo et al., 2013). One change in this adaptation is that rare cases are determined using the $\theta_{HS}$ value, instead of a relevancy function. Similarly, only high values as considered as 'rare', instead of considering both low and high values as rare, as in the original algorithm. Oversampling and undersampling are performed at rates of 400% and 0% respectively, as to obtain an equivalent number of normal and rare cases.

## 3.2 Ensemble-based techniques

Ensembles are collections of models (called individual learners), each with variations to the individual learner model type or to the training procedure (Alobaidi et al., 2019). It is well established that ensemble-based methods improve model stability and generalisability (Alobaidi et al., 2019; Brown et al., 2005). Recent advances in ensemble learning have emphasised the importance of diversity-in-learning (Alobaidi et al., 2019). Diversity can be generated both implicitly and explicitly through a variety of methods, some of which include varying the initial set of model parameters, varying the model topology, varying the training algorithm, and varying the training data (Sharkey, 1996; Brown et al., 2005). The largest source of diversity in the ensembles under study is attributable with varying the training data, which occurs both in the various resampling methods described above and the in some cases, the ensemble algorithms. Only homogeneous ensembles are used in this work, thus no diversity is obtained through varying the model topology or training algorithm (Zhang et al., 2018; Alobaidi et al., 2019). Ensemble predictions are combined to form a single discrete prediction. Ensembles that are combined to produce discrete

predictions have been proven to outperform single models by reducing model bias and variance, thus improving overall model generalisability (Brown et al., 2005; Sharkey, 1996; Shu and Burn, 2004; Alobaidi et al., 2019). This has contributed to their widespread application in hydrological modelling (Abrahart et al., 2012). In some cases, ensembles are not combined, and the collection of predictions are used to estimate the uncertainty associated with the diversity between ensemble members (Tiwari and Chatterjee, 2010; Abrahart et al., 2012). While this approach has obvious advantages, it is not possible for all types of ensembles, such as the boosting methods, which are also used in this research. Thus, this research combines ensembles to aid comparison across the different resampling and ensemble methods used.

There are many distinct methods for creating ensemble methods. The purpose of this paper is not to review all ensemble algorithms, but rather to compare four ensemble methods that commonly appear in literature: Bagging, adaptive boosting, and gradient boosting. A fourth method, randomised weights and biases, which does not qualify as an ensemble technique due to the absence of repeated resampling, is also included in the ensemble comparison because of its widespread use. While several studies have provided comparisons of ensemble methods, none of these studies have explicitly studied their effects on high stage prediction, nor their combination with resampling strategies, which is common in applications outside of flow forecasting.

Methods that aim to improve generalisability have shown promise in achieving improved prediction on high stage, which may be scarcely represented in training data. However, to the knowledge of the authors, no research has explicitly evaluated the efficacy of ensemble-based methods for improving high stage accuracy. Applications of ensemble methods for improving performance of imbalanced target variables have been thoroughly studied in classification literature. Several classification studies have demonstrated how ensemble techniques can improve prediction accuracy for imbalanced classes (Galar et al., 2012; López et al., 2013; Díez-Pastor et al., 2015b, a; Błaszczyński and Stefanowski, 2015). Such methods are increasingly being adapted for regression problems, which is typically achieved by projecting continuous data into a classification dataset (Moniz et al., 2017b, a; Solomatine and Shrestha, 2004). Pseudocode for each of the ensemble algorithms used in this research is provided in Appendix B.

### 3.2.1 Randomised weights and biases

While not technically a form of ensemble learning, repeatedly randomising the weights and biases of ANNs is one of the simplest and most common methods for achieving diversity among a collection of models, thus, it acts as a good comparison point for the proceeding ensemble methods (Brown et al., 2005). In this method, members are only distinguished by the randomisation of the initial parameter values (i.e., the initial weights and biases for ANNs in this research) used for training. For this method, an ensemble of ANNs is trained, each member having a different randomised set of initial weights and biases. Thus when trained, each ensemble member may converge to different final weight and bias values. Ensemble members are combined through averaging. This technique is often used, largely to alleviate variability in training outcomes and uncertainty associated with the initial weight and bias parameterisation (Shu and Burn, 2004; de Vos and Rientjes, 2005; Fleming et al., 2015; Barzegar et al., 2019). Despite its simplicity, this method has been demonstrated to produce considerable improvements in performance when compared to a single ANN model, even outperforming more complex ensemble methods (Shu and Burn,

2004). The weights and biases of each ANN are initialised using the default initialisation function in MATLAB and an ensemble size of 20 is used.

### 3.2.2 Bagging

Bagging is a widely used ensemble method first introduced in (Breiman, 1996). Bagging employs the bootstrap resampling method, which consists of sampling with replacement, to generate subsets of data on which to train ensemble members. The ensemble members are combined through simple averaging to form discrete predictions. Bagging is a proven ensemble method in flood prediction studies and has been widely applied and refined for, both spatial and temporal prediction, since its introduction by Breiman (1996). Chapi et al. (2017) use Bagging with Logistic Model Trees (LMT) as the individual learners to predict spatial flood susceptibility. The Bagging ensemble is found to outperform standalone LMTs, in addition to logistic regression and Bayesian logistic regression. For a similar flood susceptibility prediction application, Chen et al. (2019) use Bagging with Reduced Error Pruning Trees (REPTree) as the base learners. The Bagged models are compared to Random Subspace ensembles; both ensemble methods perform better than the standalone REPTree models, with the Random Subspace model slightly outperforming the Bagged ensemble. Anctil and Lauzon (2004) compared five generalisation techniques in the development of ANNs for flow forecasting. They combined Bagging, boosting and stacking with stop training and Bayesian regularisation, making a total of nine model configurations. They found that stacking, Bagging, and boosting all resulted in improved model performance, ultimately recommending the use of the last two in conjunction with either stop training or Bayesian regularisation. Ouarda and Shu (2009) compared stacking and Bagging ANN models against parametric regression for estimating low flow quantile for summer and winter seasons and found higher performance in ANN models (single and ensemble) compared to traditional regression models (Ouarda and Shu, 2009). Cannon and Whitfield (2002) applied Bagging to MLP-ANN models for predicting flow and found that Bagging helped create the best performing ensemble ANN. Shu and Burn (2004) evaluated six approaches for creating ANN ensembles for regional flood frequency flood analysis, including Bagging combined with either simple averaging or stacking; Bagging resulted in higher performance compared to the basic ensemble method. In a later study, Shu and Ouarda (2007) used Bagging and simple averaging to create ANN ensembles for estimating regional flood quantiles at ungauged sites. Implementing Bagging is uncomplicated, a description of the algorithm is described in its original appearance (Breiman, 1996). This research uses a Bagging ensemble of 20 members.

### 3.2.3 Adaptive boosting for regression

The AdaBoost algorithm was originally developed by Freund and Schapire (1996) for classification problems. The algorithm has undergone widespread adaptation and its popularity has lead to the development of many variations, which typically introduce improvements in performance, efficiency, and expanded for regression problems. This study uses the AdaBoost.RT variation (Solomatine and Shrestha, 2004; Shrestha and Solomatine, 2006). Broadly put, the AdaBoost algorithm begins by training an initial model. The following model in the ensemble is trained using a resampled or reweighted training set, based on the residual error of the previous model. This process is typically repeated until the desired ensemble size is achieved or a

stopping criterion is met. Predictions are obtained by weighted combination of the ensemble members, where model weights are a function of their overall error.

Similar to Bagging, there are many examples of AdaBoost applications for hydrological prediction. Solomatine and Shrestha (2004) compared various forms of AdaBoost against Bagging in models predicting river flows and found AdaBoost.RT to outperform Bagging. In a later study, the same authors compared the performance of AdaBoosted M5 tree models against ANN models for various applications, including predicting river flows in a catchment; they found higher performance in models that used the AdaBoost.RT algorithm compared to single ANNs (Shrestha and Solomatine, 2006). Liu et al. (2014) used AdaBoost.RT for calibrating process-based rainfall-runoff models, and found improved performance over the single model predictions. Wu et al. (2020) compared boosted ensembles against Bagged ensembles for predicting hourly streamflow and found the combination of AdaBoost (using resampling) and Bayesian model averaging gave the highest performance.

The variant of AdaBoost in this research follows the algorithm AdaBoost.RT proposed by (Solomatine and Shrestha, 2004; Shrestha and Solomatine, 2006). This algorithm has three hyperparameters. The relative error threshold parameter is selected as the 80th percentile of the residuals of the individual learner and 20 ensemble members are trained. AdaBoost can be performed using either resampling or reweighting (Shrestha and Solomatine, 2006); resampling is used in this research as it has been found to typically outperform reweighting (Seiffert et al., 2008). Recently, several studies have independently proposed a modification to the original AdaBoost.RT algorithm by adaptively calculating the relative error threshold value for each new ensemble member (Wang et al., 2019b; Li et al., 2020). This modification to the algorithm was generally found to be detrimental to the performance of the models in the present research, thus, the static error threshold described in the original algorithm description was used (Solomatine and Shrestha, 2004).

### 3.2.4 Least squares boosting

LSBoost is a variant of gradient boosting, which is an algorithm that involves training an initial model, followed by a sequence of models that are each trained to predict the residuals of the previous model in the sequence. This is in contrast to the AdaBoost method, which uses the model residuals to inform a weighted sampling scheme for subsequent models. The prediction at a given training iteration is calculated by the weighted summation of the already trained model(s) from the previous iterations. For LSBoost weighting is determined by a least-squares loss function; other variants of gradient boosting use a different loss function (Friedman, 2000).

Gradient boosting algorithms have previously been used to improve efficiency and accuracy for hydrological forecasting applications. Ni et al. (2020) use the gradient boosting variant XGBoost, which uses Desision Trees (DTs) as the individual learners, in combination with a Gaussian Mixture Model (GMM) for streamflow forecasting. The GMM is used to cluster streamflow data, and an XGBoost ensemble is fit to each cluster. Clustering streamflow data into distinct subsets for training is sometimes used as an alternative to resampling; its purpose is similar to that of resampling, which is to change the training sample distribution (Wang et al., 2006). The combination of XGBoost and GMM is found to outperform standalone SVM models. Erdal and Karakurt (2013) developed gradient boosted regression trees and ANNs for predicting daily streamflow and found gradient boosted ANNs to have higher performance than the regression tree counterparts. Worland et al. (2018) use

gradient boosted regression trees to predict annual minimum 7-day streamflow at 224 unregulated sites; performance is found to be competitive with several other types of data-driven models. Zhang et al. (2019) use the Online XGBoost gradient boosting algorithm for regression tree models to simulate streamflow and found that it outperformed many other data-driven and lumped hydrological models. Papacharalampous et al. (2019) use gradient boosting with regression trees and linear models, which are compared against several other model types for physically-based hydrological model quantile regression post-processing. Neither of the gradient boosting models outperform the other regression models and a uniformly weighted ensemble of all other model types typically outperforms any individual model type. These examples of gradient boosting affirm its capability for improving performance compared to the single model comparison as well as other machine learning models. However, none of these studies use gradient boosting with ANNs as the individual learner. Moreover, these studies do not examine the effects of gradient boosting on model behaviour within the context of the imbalance problem. Therefore, we use LSBoost to study its efficacy for improving high stage performance.

The implementation of LSBoost in this research is unchanged from the original algorithm (Friedman, 2000). The algorithm has two hyperparameters; the learning rate which scales the contribution of each new model and the number of boosts. A learning rate of 1 is used and the number an ensemble size of 20 is used.

## 3.3 Hybrid methods

The resampling and training strategies reviewed above can be combined to further improve model performance on imbalanced data; numerous algorithms have been proposed in literature that embed resampling schemes in ensemble learning methods. Galar et al. (2012) describes a taxonomy and presents a comprehensive comparison of such algorithms for classification problems. Many of these algorithms effectively present minor improvements or refinements to popular approaches. Alternative to implementing every single unique algorithm for training ensembles, the present research proposes employing a systematic approach to combine preprocessing resampling and ensemble training algorithms, in a modular fashion; such combinations are referred to as 'hybrid methods'. Hybrid methods hope to achieve the benefits of both standalone methods: improved performance on high stage while maintaining good generalisability. Thus, in this research, every permutation of resampling (RUS, ROS, and SMOTER) and ensemble methods (RWB, Bagging, AdaBoost, and LSBoost) is evaluated, resulting in twelve unique hybrid methods. For resampling combinations with RWB ensembles, the resampling is performed once, thus, diversity is only obtained from the initialisation of the ANN. This combination is equivalent to evaluating each resampling technique individually, to provide a basis for comparison with resampling repeated for each ensemble member, as used in the other ensemble-based configurations. For combinations of resampling with Bagging, AdaBoost, and LSBoost, the resampling procedure is performed for training each new ensemble member. One non-intuitive hybrid case is the combination of SMOTER with AdaBoost, because the synthetically generated samples do not have predetermined error weights. A previous study has recommended assigning the initial weight value to synthetic samples (Díez-Pastor et al., 2015a). However, this research proposes that synthetic sample weights are calculated in the same manner as the synthetic samples (e.g., based on the randomly interpolated point between a sample and a random neighbouring point). Thus, if two samples with relatively high weights are used to generate a synthetic sample, the new sample will have a similar weight.

**Table 4.** Summary of ensemble methods and hyperparameters.

| Type | Complete name | Short form | Hyperparameters |
|------|---------------|------------|-----------------|
| Resampling | Random undersampling | RUS | Rare case threshold ($\theta_{HS}$) = 80th percentile stage |
| | Random oversampling | ROS | Rare case threshold ($\theta_{HS}$) = 80th percentile stage |
| | Synthetic minority oversampling technique | SMOTER | Rare case threshold ($\theta_{HS}$) = 80th percentile stage<br>Oversampling percentage = 400%<br>Undersampling percentage = 0%<br>K-nearest neighbours = 10 |
| Ensemble | Randomized initial weights and biases | RWB | - |
| | Bootstrap aggregating | Bagging | Combination weighting: uniform |
| | Adaptive boosting (for regression using error thresholding) | AdaBoost | Error threshold = 80th percentile of base model error<br>Resampling/reweighting= resampling |
| | Least squares boosting | LSBoost | Learning rate = 1<br>Combination weight = least squares |

The hyperparameters for each of the resampling and ensemble method employed in this study are listed in Table 4. Every
ensemble uses the ANN described in Sect. 2.2 as the individual learner. The hyperparameters of the individual learner are kept
the same throughout all of the ensemble methods to allow for a fair comparison (Shu and Burn, 2004) (excluding of course the
number of epochs, which is determined through validation stop-training).

### 3.4 Model implementation and evaluation

All aspects of this work are implemented in MATLAB 2020a. The Neural Network Toolbox was used to train the baseline
ANN models. The resampling and ensemble algorithms used in this research were programmed by the authors and available
upon request; the pseudocode for each method is available in Appendix B.

#### 3.4.1 Performance assessment

The challenges of training models on imbalanced datasets outlined in Sect. 1 and evaluating model performance are one and
the same: many traditional performance metrics (e.g., MSE, $CE$, etc.) are biased towards the most frequent stage values and
the metrics are insensitive to changes in high stage accuracy. In fact, despite their widespread use, these metrics are criticised in
literature. For example, ANN models for sunspot prediction produced a lower RMSE (equivalent to $CE$ when used on datasets
with the same observed mean) compared to conventional models, however were found to have no predictive value (Abrahart
et al., 2007). Similarly, $CE$ values may be misleadingly favourable if there is significant observed seasonality (Ehret and
Zehe, 2011). $CE$ is also associated with the underestimation of peak flows, volume balance errors, and undersized variability

(Gupta et al., 2009; Ehret and Zehe, 2011). Zhan et al. (2019) suggest that $CE$ is sensitive to peak flows due to the square term. This assertion is correct while comparing two samples, however, when datasets are imbalanced, the errors of typical stage overwhelm those of high stage. Ehret and Zehe (2011) evaluate the relationship between phase error and RMSE using triangular hydrographs; their study shows how RMSE is highly sensitive to minor phase errors, however, when a hydrograph has a phase and amplitude error RMSE is much more sensitive to overpredictions compared to underpredictions.

The coefficient of efficiency (CE), commonly known as the Nash-Sutcliffe efficiency, is given by the following formula:

$$CE = 1 - \frac{\sum (q(t) - \hat{q}(t))^2}{\sum (q(t) - \bar{q})^2} \tag{1}$$

where $q$ is the observed stage, $\hat{q}$ is the predicted stage, and $\bar{q}$ is the mean observed stage.

The persistence index (PI) is a measure similar to $CE$, but instead of normalising the sum of squared error of a model based on the observed variance, it is normalised based on the sum of squared error between the target variable and itself, lagged by the lead time of the forecast model (referred to as the naive model). Thus, the $CE$ and $PI$ range from an optimum value of 1 to -$\infty$, with values of 0 corresponding to models that are indistinguishable from the observed mean and naive models, respectively. Since both models use antecedent input variables with lag times equal to the forecast length, $PI$ is a useful indicator for over-reliance on this input variable, which has been associated with peak stage timing error (de Vos and Rientjes, 2009). Furthermore, the $PI$ measure overcomes some of the weaknesses of $CE$, such as a misleadingly high value for seasonal watersheds. Moreover, $PI$ is effective in identifying when models become over-reliant on autoregressive inputs, as the model predictions will resemble those of the naive model. $PI$ is given by the following formula:

$$PI = 1 - \frac{\sum (q(t) - \hat{q}(t))^2}{\sum (q(t) - q(t - L))^2} \tag{2}$$

where $L$ is the lead time of the forecast.

In order to quantify changes in model performance on high stage, both the $CE$ and $PI$ measures are calculated for typical stage ($TS$) and high stage ($HS$) (Crochemore et al., 2015). The resampling methods are expected to improve the high stage $CE$ at the expense of $CE$ for typical stage, while ensemble methods are expected to produce an outright improvement in model generalisation, reflected by reduced loss in performance between the calibration and test data partitions. Thus, the objective of this research is to find model configurations with improved performance on high stage while maintaining strong performance overall. $TS$ and $HS$ performance metrics are calculated based only on the respective observed stage. For example, the $CE$ for high stage is calculated by:

$$CE_{HS} = 1 - \frac{\sum (q_{HS}(t) - \hat{q}_{HS}(t))^2}{\sum (q_{HS}(t) - \bar{q}_{HS})^2} \tag{3}$$

where $q_{HS}$ is given by:

$$q_{HS} = q \,|\, q \geq \theta_{HS} \tag{4}$$

The performance for $CE_{TS}$, $PI_{HS}$, and $PI_{TS}$ are calculated in the same manner, substituting $q_{TS}(t)$ for $q_{HS}(t)$ in Eq. 4 for $HS$ calculations, and using Eq. 2 in place of Eq. 1 for $PI$ calculations.

### 3.4.2 K-fold cross-validation

The entire available dataset is used for both training and testing by the use of KFCV, a widely used cross-validation method (Hastie et al., 2009; Bennett et al., 2013; Solomatine and Ostfeld, 2008; Snieder et al., 2020). Ten folds are used in total; eight folds for calibration and two for testing. Of the eight calibration folds, six are used for training while two are used for early-stopping. When performance is reported as a single value, it refers to the mean model performance of the respective partition across K-folds. It is important to distinguish between the application of KFCV for evaluation (as used in this research) as opposed to using KFCV for producing ensembles, in which an ensemble of models is trained based on a KFCV data partitioning scheme (Duncan, 2014).

## 4 Results

This section provides a comparison of the performance of each of the methods described throughout Sect. 3 applied to the Bow and Don River watersheds, which are described in Sect. 2.1. Changes to model performance are typically discussed relative to the individual learner (see Sect. 2.2), unless explicit comparisons are specified. First, the results of a grid-search analysis of ensemble size is provided. Next, general overview and comparison of the results are presented, followed by detailed comparison of the resampling and ensemble methods. Finally, the effects that varying the $HS$ threshold and ensemble size have on resampling and high stage performance are evaluated for the Bagging and SMOTER-Bagging models.

Fig. 6 illustrates the change in test performance as the ensemble size increases from 2 to 100 for each river. This grid-search is performed only for the base ensemble methods (RWB, Bagging, AdaBoost, and LSBoost) without any resampling. The Bow River results indicates that AdaBoost and LSBoost tend to favour a small ensemble size (2-15 members), whereas the generalisation of RWB and Bagging improves with a larger size (>20 members). The performance of LSBoost rapidly deteriorates as the ensemble size grows, likely as the effects of overfitting become more pronounced. Similar results are obtained for the Don, except that RWB, Bagging, and AdaBoost all improve with larger ensemble size, while LSBoost performs worse than all other ensembles, even for small ensemble sizes. Similar to the Bow, a larger ensemble size (>20 members) produces favourable MSE.

Figs. 7 and 8 show the $CE$ and $PI$ box-whisker plots for the Bow and Don Rivers, respectively. These figures show the performance of the test dataset, across the K-folds, for each resampling, ensemble, and hybrid technique, as well as the individual learner. The performance metrics are calculated for the entire dataset, the $HS$ values, and the $TS$ values. Models with a larger range have more variable performance when evaluated across different subsets of the available data.

The average performance for each resampling, ensemble, and hybrid methods for the Bow and Don River models are shown in Tables 5 and 6, respectively, which list the $CE$ and $PI$ for the entire dataset, as well as the $TS$ and the $HS$ datasets. The ensemble results for each KFCV fold were combined using a simple arithmetic average. The results have been separated into different categories: each section starts with the ensemble technique (either RWB, Bagging, AdaBoost, or LSBoost), followed by the three hybrid variations (RUS-, ROS-, or SMOTER-). The calibration (training and validation) performance is indicated in parentheses and italics, followed by the test performance. Comparing both the calibration and test performance is useful

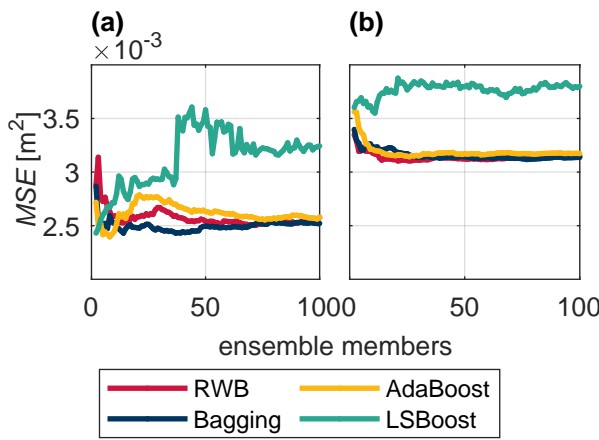

**Figure 6.** Test $MSE$ across ensemble size for RWB (red), Bagging (blue), AdaBoost (yellow), and LSBoost (green) for the Don (a) and Bow River (b).

since it provides a sense of overfitting, hence, generalisation. For example, an improvement in calibration performance and decrease in test performance suggests that the model has been overfitted. In contrast, improvements to both partitions indicates favourable model generalisation. The best performing model (based on testing performance) have been highlighted in bold text for each performance metric, $CE$ and $PI$, for both watersheds.

Based on the $CE$ values in Figs. 7 - 8 and Tables 5 - 6, the majority of the Bow and Don River models achieve "acceptable" prediction accuracy (as defined by Mosavi et al. (2018)). Values of $CE_{TS}$ and $CE_{HS}$ are both lower than the $CE$, which is to be expected as the stage variance of each subset is lower than that of the the set of all stage values. For the Bow River models, the $CE$ and $CE_{TS}$ values are consistently higher than the $CE_{HS}$; this is attributable to the high seasonality of the watershed producing a misleadingly high value for $CE$ due to the high variance of stage throughout the year, as discussed in Sect. 3.4.1. The $CE_{HS}$ values also have higher variability compared to the overall $CE$ and $CE_{TS}$, as shown in Fig. 7a. In contrast, for the Don River models, the difference in $CE$, $CE_{TS}$, and $CE_{HS}$ is less pronounced; whereas the $CE$ (for the entire dataset) is typically higher, as expected, than both the $CE_{TS}$ and $CE_{HS}$, the difference between $CE_{TS}$ and $CE_{HS}$ is low, as demonstrated in the mean and range of the box-whisker plots in Fig. 8a. Unlike the Bow River, the Don River does not exhibit notable seasonality, resulting in smaller difference between the $HS$ and $TS$.

Values of $PI$ are typically lower than for $CE$ for both watersheds. The Bow River models obtain $PI$ values centred around 0 (see Fig. 7b), indicating that only some of the model configurations perform with greater accuracy than the naive model, meaning that a timing error exists. The box-whisker plots of each ensemble method do not show a clear trend (with respect to the mean value or range) when comparing the $PI$, $PI_{TS}$, and $PI_{HS}$: the mean and range are similar for all variants tested.

The Don River models have positive $PI$ values of approximately 0.6, indicating a lower reliance on autoregressive input variables, when compared to the Bow River. And in contrast to the Bow River, there is a notable difference between the $PI$ metrics: the $PI_{TS}$ has a lower mean value and higher variance (see Fig. 8b) than the $PI$ (for the entire dataset) and the $PI_{HS}$.

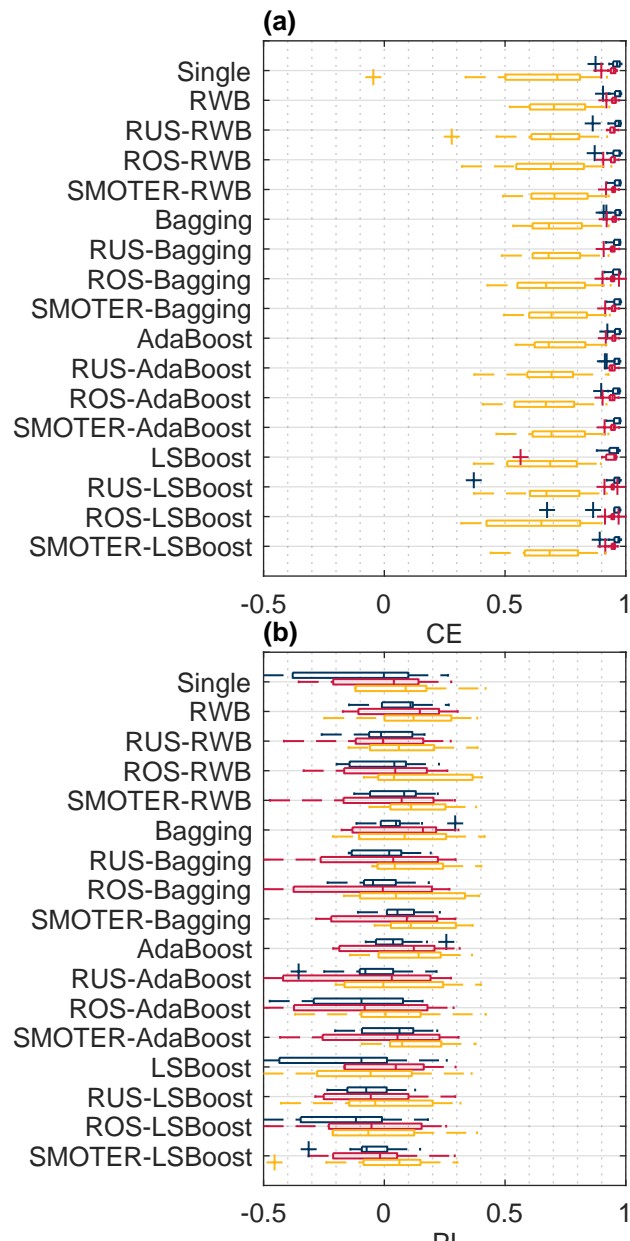

**Figure 7.** Overall (blue), typical stage (red), and high stage (yellow) $CE$ (a) and $PI$ (b) for the Bow River models.

These lower $PI_{TS}$ are due to the low variability (steadiness) of the Don River $TFs$ (see Fig. 4), and thus, the sum of squared error between the naive model and observed stage is also low, reducing the $PI$ value. The low value of $PI_{TS}$ is attributed to the quality of the naive model, not the inaccuracy of the ANN counterpart. Note that $PI_{HS}$ are typically slightly higher than

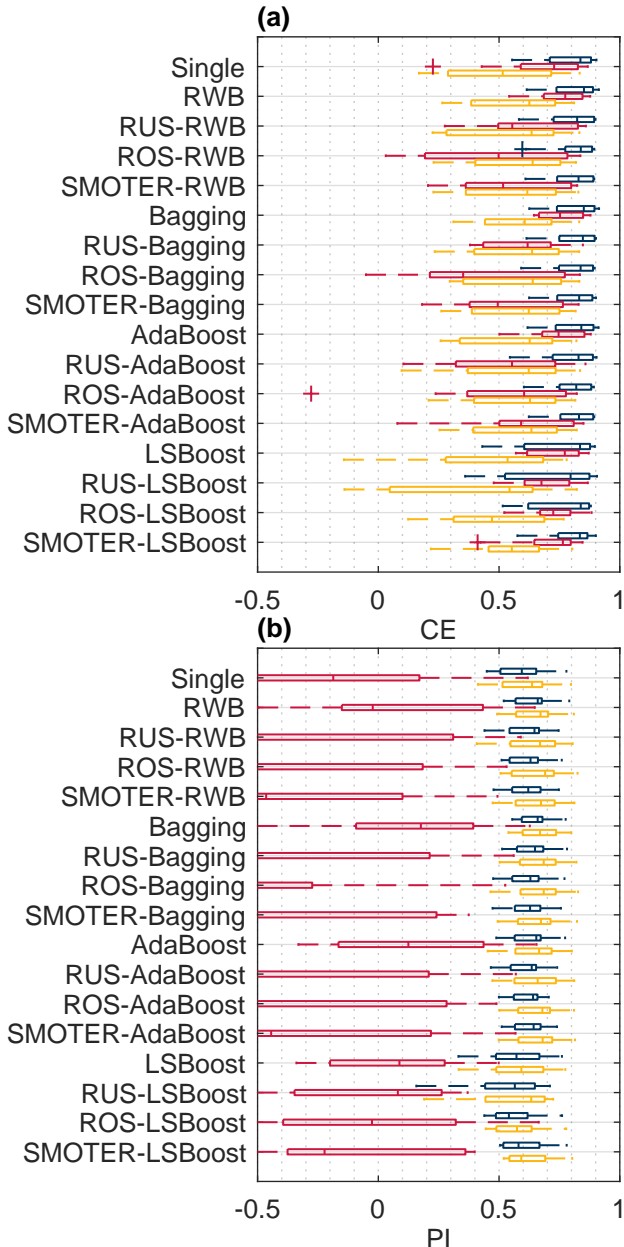

**Figure 8.** Overall (blue), typical stage (red), and high stage (yellow) $CE$ (a) and $PI$ (b) for the Don River models.

the overall PI: during high stage, there is greater variability, thus the naive model is less accurate, resulting in a higher $PI$ score.

**Table 5.** Mean $CE$ and $PI$ scores for all, typical, and high stage for the Bow River ensembles; the highest scores are shown in bold and the calibration scores are italicised and enclosed by parentheses.

| Label | CE | CE$_{TS}$ | CE$_{HS}$ | PI | PI$_{TS}$ | PI$_{HS}$ |
|---|---|---|---|---|---|---|
| Base model | *(0.967)* 0.954 | *(0.954)* 0.944 | *(0.829)* 0.617 | *(0.182)* -0.166 | *(0.111)* -0.0593 | *(0.227)* -0.175 |
| RWB | *(0.974)* 0.962 | *(0.96)* 0.951 | *(0.865)* 0.718 | *(0.331)* **0.0731** | *(0.229)* 0.0856 | *(0.392)* 0.128 |
| RUS-RWB | *(0.972)* 0.956 | *(0.954)* 0.947 | *(0.863)* 0.68 | *(0.286)* -0.0505 | *(0.116)* -0.013 | *(0.384)* 0.015 |
| ROS-RWB | *(0.973)* 0.957 | *(0.955)* 0.947 | *(0.87)* 0.681 | *(0.312)* -0.0266 | *(0.125)* 0.00468 | *(0.418)* 0.0454 |
| SMOTER-RWB | *(0.974)* **0.963** | *(0.957)* 0.948 | *(0.871)* **0.72** | *(0.329)* 0.0524 | *(0.176)* 0.0168 | *(0.417)* 0.139 |
| Bagging | *(0.973)* 0.961 | *(0.96)* **0.952** | *(0.86)* 0.709 | *(0.32)* 0.0503 | *(0.234)* **0.0886** | *(0.372)* 0.0887 |
| RUS-Bagging | *(0.972)* 0.961 | *(0.955)* 0.945 | *(0.867)* 0.715 | *(0.298)* 0.00346 | *(0.119)* -0.0403 | *(0.399)* 0.116 |
| ROS-Bagging | *(0.973)* 0.959 | *(0.954)* 0.943 | *(0.873)* 0.696 | *(0.312)* -0.0374 | *(0.111)* -0.0851 | *(0.425)* 0.0896 |
| SMOTER-Bagging | *(0.974)* 0.962 | *(0.957)* 0.948 | *(0.873)* 0.719 | *(0.333)* 0.0511 | *(0.17)* 0.018 | *(0.427)* **0.144** |
| AdaBoost | *(0.974)* 0.963 | *(0.96)* 0.95 | *(0.865)* 0.719 | *(0.327)* 0.0465 | *(0.22)* 0.0488 | *(0.389)* 0.112 |
| RUS-AdaBoost | *(0.972)* 0.959 | *(0.954)* 0.942 | *(0.865)* 0.693 | *(0.288)* -0.0642 | *(0.107)* -0.105 | *(0.39)* 0.0509 |
| ROS-AdaBoost | *(0.972)* 0.956 | *(0.951)* 0.942 | *(0.872)* 0.673 | *(0.291)* -0.114 | *(0.052)* -0.109 | *(0.424)* -0.0307 |
| SMOTER-AdaBoost | *(0.974)* 0.962 | *(0.957)* 0.947 | *(0.872)* 0.714 | *(0.331)* 0.0259 | *(0.166)* -0.00642 | *(0.425)* 0.121 |
| LSBoost | *(0.974)* 0.948 | *(0.958)* 0.907 | *(0.869)* 0.666 | *(0.328)* -0.504 | *(0.189)* -0.786 | *(0.403)* -0.104 |
| RUS-LSBoost | *(0.97)* 0.904 | *(0.952)* 0.944 | *(0.854)* 0.364 | *(0.246)* -0.718 | *(0.0643)* -0.0609 | *(0.35)* -0.824 |
| ROS-LSBoost | *(0.973)* 0.929 | *(0.952)* 0.944 | *(0.875)* 0.517 | *(0.304)* -0.425 | *(0.0638)* -0.0757 | *(0.435)* -0.431 |
| SMOTER-LSBoost | *(0.973)* 0.958 | *(0.954)* 0.946 | *(0.868)* 0.684 | *(0.3)* -0.0522 | *(0.117)* -0.0255 | *(0.401)* 0.00239 |

## 4.1 Comparison of resampling and ensemble methods

This section provide a detailed comparison of performance across the different resampling and ensemble methods. As expected, all three resampling methods (RUS, ROS,and SMOTER) typically increase $HS$ performance, often at the expense of $TS$ performance. Based on results shown in Table 5, the SMOTER- variations provide the highest performance for $HS$ for the Bow River. SMOTER-RWB $CE_{HS}$ is 0.72, an increase from 0.617 of the individual learner, whereas the SMOTER-Bagging $PI_{HS}$ is 0.144, compared to -0.175 for the individual learner. These indicators suggest that the $HS$ prediction accuracy has improved slightly using these SMOTER variations. The results shown in Table 6 for the Don River indicate that the best improvements for $HS$ prediction accuracy is provided by the RUS-Bagging method: the $CE_{HS}$ is 0.585 (an increase from 0.511 of the individual learner), and the $PI_{HS}$ is 0.668 (an increase from 0.61 of the individual learner). While both these metrics show an improvement in $HS$ prediction accuracy for the Don River, the improvements are relatively small compared to the performance improvement for the Bow River. ROS often exhibits poorer performance than SMOTER and RUS. Previous research has noted the tendency for ROS-based models to overfit, due to the high number of duplicate samples (Yap et al., 2014). RUS, despite using considerable less training data for each individual learner, is not as prone to overfitting as ROS.

**Table 6.** Mean $CE$ and $PI$ scores for all, typical, and high stage for the Don River ensembles; the highest scores are shown in bold and the calibration scores are italicised and enclosed by parentheses

| Label | CE | CE$_{TS}$ | CE$_{HS}$ | PI | PI$_{TS}$ | PI$_{HS}$ |
|---|---|---|---|---|---|---|
| Base model | *(0.86)* 0.781 | *(0.782)* 0.664 | *(0.677)* 0.511 | *(0.716)* 0.592 | *(0.0197)* -0.213 | *(0.74)* 0.61 |
| RWB | *(0.873)* 0.806 | *(0.814)* 0.755 | *(0.705)* 0.572 | *(0.744)* 0.641 | *(0.165)* **0.0944** | *(0.763)* 0.654 |
| RUS-RWB | *(0.853)* 0.792 | *(0.638)* 0.588 | *(0.685)* 0.555 | *(0.704)* 0.615 | *(-0.585)* -0.63 | *(0.746)* 0.645 |
| ROS-RWB | *(0.864)* 0.799 | *(0.629)* 0.488 | *(0.715)* 0.584 | *(0.726)* 0.624 | *(-0.632)* -0.991 | *(0.771)* 0.665 |
| SMOTER-RWB | *(0.866)* 0.795 | *(0.642)* 0.552 | *(0.715)* 0.57 | *(0.729)* 0.618 | *(-0.573)* -0.749 | *(0.771)* 0.656 |
| Bagging | *(0.869)* **0.808** | *(0.811)* **0.757** | *(0.696)* 0.581 | *(0.736)* **0.65** | *(0.154)* 0.0875 | *(0.755)* 0.663 |
| RUS-Bagging | *(0.864)* 0.805 | *(0.676)* 0.609 | *(0.706)* **0.585** | *(0.726)* 0.638 | *(-0.433)* -0.502 | *(0.764)* **0.668** |
| ROS-Bagging | *(0.858)* 0.795 | *(0.553)* 0.271 | *(0.716)* 0.584 | *(0.712)* 0.618 | *(-1.14)* -1.41 | *(0.771)* 0.665 |
| SMOTER-Bagging | *(0.865)* 0.798 | *(0.604)* 0.526 | *(0.718)* 0.581 | *(0.729)* 0.623 | *(-0.705)* -0.888 | *(0.774)* 0.662 |
| AdaBoost | *(0.87)* 0.803 | *(0.807)* 0.744 | *(0.698)* 0.567 | *(0.737)* 0.637 | *(0.136)* 0.0393 | *(0.758)* 0.651 |
| RUS-AdaBoost | *(0.857)* 0.787 | *(0.658)* 0.53 | *(0.694)* 0.553 | *(0.712)* 0.613 | *(-0.51)* -0.888 | *(0.754)* 0.646 |
| ROS-AdaBoost | *(0.864)* 0.793 | *(0.604)* 0.516 | *(0.718)* 0.575 | *(0.726)* 0.616 | *(-0.725)* -1.07 | *(0.773)* 0.658 |
| SMOTER-AdaBoost | *(0.867)* 0.801 | *(0.667)* 0.578 | *(0.715)* 0.584 | *(0.732)* 0.629 | *(-0.46)* -0.743 | *(0.771)* 0.665 |
| LSBoost | *(0.869)* 0.746 | *(0.813)* 0.741 | *(0.696)* 0.446 | *(0.736)* 0.555 | *(0.169)* 0.0719 | *(0.755)* 0.567 |
| RUS-LSBoost | *(0.835)* 0.715 | *(0.744)* 0.685 | *(0.625)* 0.419 | *(0.67)* 0.513 | *(-0.128)* -0.207 | *(0.697)* 0.548 |
| ROS-LSBoost | *(0.871)* 0.759 | *(0.761)* 0.716 | *(0.708)* 0.472 | *(0.738)* 0.561 | *(-0.0738)* -0.0931 | *(0.766)* 0.579 |
| SMOTER-LSBoost | *(0.871)* 0.787 | *(0.775)* 0.695 | *(0.707)* 0.537 | *(0.74)* 0.599 | *(0.00723)* -0.0914 | *(0.765)* 0.62 |

The RUS-Bagging models consistently outperform the RUS-RWB models; this may be due to the repeated resampling, thus RUS-Bagging uses much more of the original training samples, while RUS-RWB only uses 20% of the original data.

Figures 9 and 10 show absolute changes in $CE$ and $PI$ relative to the individual learner for the Bow and Don Rivers, respectively, for the entire dataset, the $TS$ and the $HS$. Performance is colourised in a 2D matrix to facilitate comparisons in performance between each resampling methods across ensemble types and vice versa. From these figures, it is apparent that SMOTER generally produces the largest improvements in $HS$ performance, for both $CE$ and $PI$, and for both watersheds. The SMOTER methods are also generally the least detrimental to $TS$ performance for both watersheds, as compared to ROS and RUS. Notably, SMOTER is the only resampling method whose performance does not decrease when used in combination with LSBoost. However, the change in performance due to SMOTER is marginal compared to the models without resampling. For the Bow River, the largest improvements between the best models with no resampling and the best models with resampling for $CE_{HS}$ and $PI_{HS}$ are 0.001 and 0.016, respectively. For the Don River, the same improvements are 0.004 and 0.005, respectively. The remaining resampling methods (RUS and ROS) also generally tend to improve $HS$ performance across the ensemble techniques; however this improvement is not consistent, as is the case with SMOTER, and the decrease is $TS$

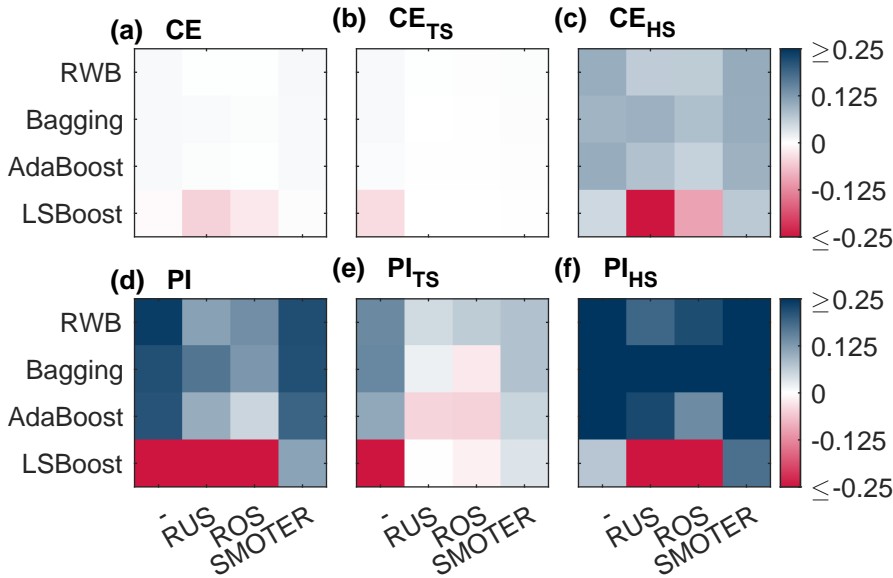

**Figure 9.** Change in (absolute) performance of $CE$ (a), $CE_{TS}$ (b), $CE_{HS}$ (c), $PI$ (d), $PI_{TS}$ (e), $PI_{HS}$ (f) produced by combinations of resampling (listed along the x-axis) and ensemble (listed along the y-axis) methods for the Bow River models.

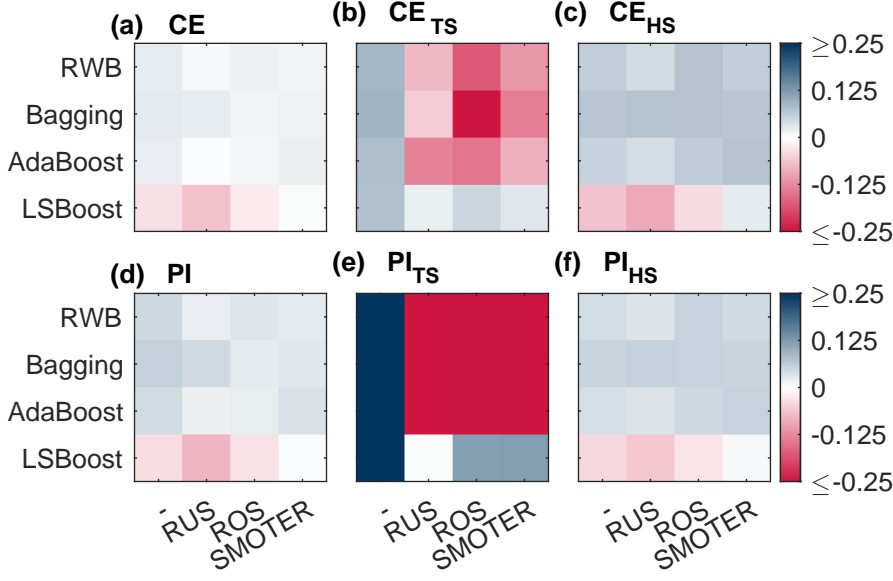

**Figure 10.** Change in (absolute) performance of $CE$ (a), $CE_{TS}$ (b), $CE_{HS}$ (c), $PI$ (d), $PI_{TS}$ (e), $PI_{HS}$ (f) produced by combinations of resampling (listed along the x-axis) and ensemble (listed along the y-axis) methods for the Don River models.

performance is also higher. Thus, while SMOTER provides consistent improvements over the non-resampling methods for $CE$ and $PI$ (entire, $TS$, and $HS$), RUS and ROS only provide minor improvements to $HS$ performance.

When looking at the resampling methods, the RWB ensembles exhibit competitive performance compared to the other ensemble methods, despite their lower diversity. These ensembles represent a considerable improvement over the individual learner and often achieve higher performance compared to the other, more complex ensemble methods, as shown in Tables 5 and 6. This suggests that using RWB is useful for improving $CE$ and $PI$ performance (for entire, $TS$, and $HS$) as compared to the single, individual learner. For the Bow River, the RWB ensembles improve the $PI$ for each case (PI, $PI_{TS}$, and $PI_{HS}$), whereas only improving $CE_{HS}$. For the Don River models, a notable increase in performance is seen for both $CE$ and $PI$ (entire and $HS$ datasets); however, when combined with the resampling techniques (RUS, ROS, and SMOTER), the $TS$ performance metrics exhibit poorer performance.

The Bagging ensembles also perform well, typically outperforming the RWB counterparts, and following the same trends described above. This is likely due to their repeated resampling, which achieves greater ensemble diversity compared to the RWB models, for which resampling only occurs once. This result is consistent with a previous comparison of Bagging and boosting (Shu and Burn, 2004). Like RWB and Bagging, AdaBoost improves model performance compared to the individual learner, but is typically slightly poorer compared to RWB and Bagging, and has higher variability in terms of improvement to model performance across all model types and both watersheds. The RWB, Bagging, and Adaboost models consistently improve $TS$ and $HS$ performance compared to the individual learner regardless of whether they are combined with a resampling strategy.

The LSBoost models have the poorest $HS$ performance out of all the ensemble methods studied. This is consistent across all resampling methods and both watersheds. In contrast, the change in performance for $CE_{TS}$ and $PI_{TS}$ is less detrimental when using LSBoost, suggesting that this method is not well-suited to improve $HS$ performance. The LSBoost models are slightly overfitted, despite utilising the stop-training for calibrating the ANN ensemble members. This is indicated by the degradation in performance between the calibration and test dataset, a change which is larger than that seen in the other ensemble models. This is most noticeable for the RUS-LSBoost models for both the Bow and the Don Rivers, which are more prone to overfitting compared to other models, due to the smaller number of training samples. The $CE$ decreases from 0.97 to 0.902 for the Bow and 0.835 to 0.715 for the Don River; none of the other models that use RUS exhibit such a gap between train and test performance.

The overfitting produced by the boosting methods is consistent with previous research, which finds that boosting is sometimes prone to overfitting on real-world datasets (Vezhnevets and Barinova, 2007). One reason that the improvements made by the boosting methods (AdaBoost and LSBoost) are not more substantial may be due to the use of ANNs as individual learners. ANNs typically have more degrees of freedom compared to the decision trees that are most commonly used as individual learners; thus, the additional complexity offered by boosting does little to improve model predictions. Additionally, the boosting methods further increase the effective degrees of freedom of the predictions. Nevertheless, these methods still tend to improve performance over that of the individual learner. Ensembles of less complex models such as regression trees are expected to produce relatively larger improvements when relative to the single model predictions.

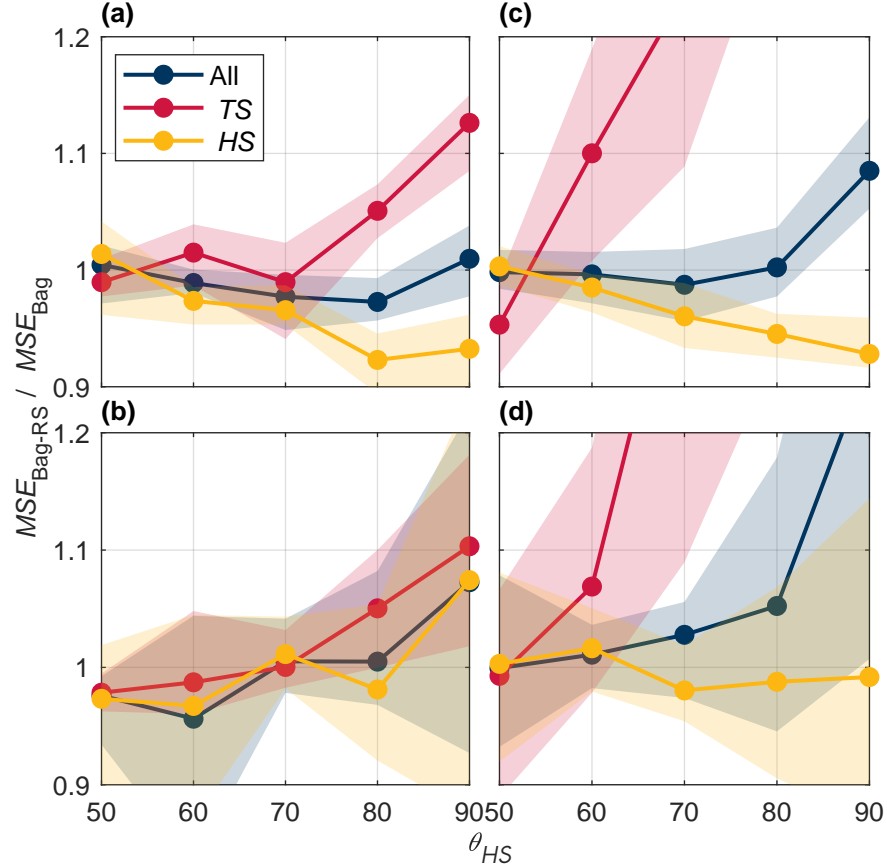

**Figure 11.** $MSE$ ratio between Bagging and SMOTER-Bagging models for the Bow River calibration (a), Bow test (c), Don River calibration (b), and Don test (d) partitions across high stage threshold values ranging from the 50th to 90th percentile stage values.

As discussed in Sect. 2.1, a fixed threshold is used to distinguish between high and typical stage values, which was set to 80% for the results presented above. Fig. 11 shows the effects of the fixed threshold increasing from the 50th to 90th percentile of the stage distribution. These plots show the relative effects of SMOTER-Bagging compared to simple Bagging; these configurations were selected for this comparison since they both exhibited relatively good, consistent performance. A performance ratio greater than 1 indicates that the SMOTER-Bagging model has greater error compared to the Bagging model, 1 indicates that they have the same performance, and less than 1, improved performance. Error is presented for all stage values as well as the $TS$ and $HS$ subsets. The calibration plots illustrate an asymmetric trade-off between $HS$ and $TS$ error. For a given $\theta_{HS}$ value, the error ratio of the $TS$ subset increases more than than the decline in $HS$ error. More importantly, the improvements in $HS$ performance obtained in calibration are considerably less pronounced in the test dataset, despite a loss in $TS$ performance.

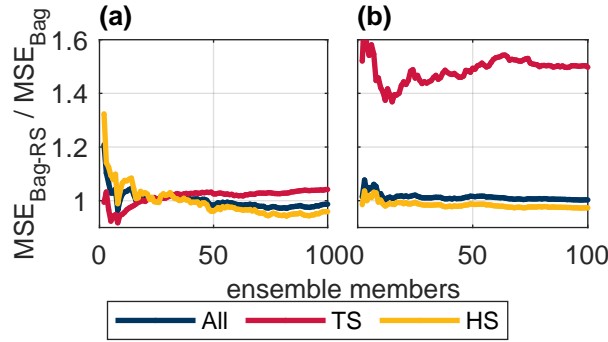

**Figure 12.** Test $MSE$ ratio between Bagging and SMOTER-Bagging models for the Bow (a) and the Don (b) Rivers across ensemble size.

Fig. 12 illustrates the effects of varying the ensemble size, thus, number of resampling repetitions, for the SMOTER-Bagging model, relative to the simple Bagging model. The plot shows the relative improvement in $HS$ produced by the SMOTER resampling as the ensemble size increases, reaching a steady value at an ensemble size of approximately 70 for both the Don and Bow systems. This is larger than that required for the simple Bagging model to reach steady performance, shown in Fig. 6, indicating that SMOTER requires more resampling than regular resampling with replacement (default in Bagging) in order to reach stable performance. Consistent with observations made from Fig. 11, an asymmetric trade-off between typical and high stage performance is noted, illustrated by disproportionate increase in error on typical stage, relative to the improvement on high stage.

### 4.2 Limitations and Future work

A limitation of this study is the lack of a systematic case-by-case hyperparameter optimisation of the models. The individual learner parameters (e.g. topology, activation function, etc.) were constant across all ensemble members. Likewise, the ensemble hyperparameters were not optimised, but simply tuned using an ad-hoc approach. A systematic approach to hyperparameter optimisation for each model will likely yield improved model performance. However, hyperparameter optimisation on such a scale would be very computationally expensive. Similarly, the selection of the $HS$ threshold may affect $CE_{HS}$ and $PI_{HS}$ performance, and the sensitivity of model performance of this threshold should be explored. This research featured resampling and ensemble methods for improving prediction accuracy across an imbalanced target dataset, i.e., the high stage. Further to imbalanced target data, flood forecasting applications commonly have imbalanced cost; for example, underprediction is typically more costly than overprediction. The use of cost-functions, such as asymmetric weighting applied to underpredictions and overpredictions, for flood forecasting has been shown to reduce underprediction of flooding (Toth, 2016). Many cost-sensitive ensemble techniques (e.g., Galar et al. (2012)) have yet to be explored in the context of flood forecasting models and should be the focus of future work.

## 5  Conclusion

This research presented the first systematic comparison of the effects of combined resampling and ensemble techniques for improving the accuracy of flow forecasting models, specifically for high stage (infrequent) observations. Methods were applied to two Canadian watersheds, the Bow River in Alberta, and the Don River, in Ontario. This research attempts to address the widespread problem of poor performance on high stage when using data-driven approaches such as ANNs. Improving performance on high stage is essential for model applications such as early flood warning systems. Three resampling and four ensemble techniques are implemented as part of ANN flow forecasting models, for both watersheds. These methods are assessed independently and systematically combined in hybrid approaches, as to assess their efficacy for improving high stage performance. A major contribution of this paper is the comprehensive evaluation of these hybrid methods, most of which are the first instances in the water resources field. While methodologies for these combination methods is available in existing machine learning literature, our proposed implementation of SMOTER-AdaBoost is a novel improvement. Results demonstrate that resampling methods, when embedded in ensemble algorithms, generally only produces a small improvement in high stage performance, based on $CE$ and $PI$; the SMOTER variation provided the most consistent improvements. An asymmetric trade-off between typical and high stage performance is observed, in which improved high stage performance produced disproportionately worse typical flow performance. Such a trade-off should be carefully considered while implementing these methods. Further research on this topic may explore the combination of cost-sensitive approaches with ensemble methods, which would allow for more aggressive penalisation of poor accuracy on high stage.

# Appendix A: Input variable selection results

**Table A1.** List of 25 most useful inputs identified using the PC IVS algorithm for the Bow and Don River watersheds, selected form the set of candidate inputs. Input variables are encoded in the following format "station ID"_"variable"_"statistic"_"lagged timesteps". Variable abbreviations "WL" and "Precip" refer to water level (stage) and precipitation.

| rank | Bow | Don |
|------|-----|-----|
| 1 | 05BH004 WL Mean L4 | HY022 WL Mean L4 |
| 2 | 05BB001 WL Max L4 | HY008 Precip Sum L4 |
| 3 | 05BB001 WL Min L12 | HY019 WL Mean L4 |
| 4 | 05BH004 WL Mean L5 | HY008 Precip Sum L5 |
| 5 | Calgary Temp Max L4 | HY027 Precip Sum L4 |
| 6 | 05BB001 WL Max L6 | HY017 WL Mean L4 |
| 7 | 05BH004 WL Mean L15 | HY022 WL Mean L5 |
| 8 | Calgary Precip Sum L5 | HY008 Precip Sum L8 |
| 9 | Calgary Temp Min L10 | HY027 Precip Sum L6 |
| 10 | Calgary Precip Sum L11 | HY017 WL Mean L5 |
| 11 | 05BH004 WL Max L4 | HY027 Precip Sum L5 |
| 12 | 05BH004 WL Min L4 | HY008 Precip Sum L10 |
| 13 | 05BH004 WL Max L7 | HY019 WL Mean L7 |
| 14 | Calgary Precip Sum L7 | HY080 WL Mean L4 |
| 15 | 05BB001 WL Min L15 | HY008 Precip Sum L11 |
| 16 | 05BH004 WL Min L8 | HY008 Precip Sum L6 |
| 17 | Calgary Precip Sum L10 | HY080 WL Mean L6 |
| 18 | 05BH004 WL Max L12 | HY027 Precip Sum L7 |
| 19 | Calgary Precip Sum L6 | HY022 WL Mean L6 |
| 20 | 05BB001 WL Max L5 | HY027 Precip Sum L8 |
| 21 | Calgary Temp Min L15 | HY022 WL Mean L7 |
| 22 | 05BH004 WL Min L6 | HY080 WL Mean L5 |
| 23 | 05BH004 WL Mean L6 | HY017 WL Mean L6 |
| 24 | 05BH004 WL Max L5 | HY080 WL Mean L7 |
| 25 | 05BB001 WL Min L9 | HY019 WL Mean L6 |

## Appendix B: Pseudocode

---

**Algorithm 1** Random undersampling

---

**Require:**

Set S containing X input features and Y observations, $(x_1, y_1), ..., (x_m, y_m)$

High stage threshold, $\theta$

$S_{TS} = S$ where $Y < \phi_{TS}$

$S_{HS} = S$ where $Y \geq \phi_{HS}$

$S'_{TS} \leftarrow sample(S_{TS}, N_{HS})$

$S'_{HS} \leftarrow sample(S_{HS}, N_{HS})$

$S' = S'_{TS} \bigcup S'_{HS}$

---

---

**Algorithm 2** Random oversampling

---

**Require:**

Set S containing X input features and Y observations, $(x_1, y_1), ..., (x_m, y_m)$

High stage threshold, $\theta$

$S_{TS} = S$ where $Y < \phi_{TS}$

$S_{HS} = S$ where $Y \geq \phi_{HS}$

$S'_{TS} \leftarrow sample(S_{TS}, N_{TS})$

$S'_{HS} \leftarrow sample(S_{HS}, N_{TS})$

$S' = S'_{TS} \bigcup S'_{HS}$

---

---

**Algorithm 3** SMOTER

---

**Require:**

Set S containing X input features and Y observations, $(x_1, y_1), ..., (x_m, y_m)$

High stage threshold, $\theta_{HS}$

**Ensure:**

$\phi_{HS}/(1 - \phi_{HS}) \, \epsilon \, \mathbb{Z}$

$N_{synth} \leftarrow \phi_{HS}/(1 - \phi_{HS}) - 1$

$S_{TS} = S$ where $Y < \phi_{TS}$

$S_{HS} = S$ where $Y \geq \phi_{HS}$

**for** $s_i \epsilon S_{HS}$ **do**

    $nn_i = \mathbf{kNN}(S, k)$

    **for** $j = 1, 2, ... N_{synth}$ **do**

        $s_j = nn_i(\mathbf{randi}(1, k))$ {randomly select one nearest neighbour}

        $s_{diff,} = s_i - s_j$

        $gap = \mathbf{rand}(0, 1)$ {randomly select a point between sample and nearest neighbour}

        $s_{synth,i,j} = s_i + s_{diff} \times gap$

    **end for**

**end for**

$S' = S \bigcup S_{synth}$ {merge original and synthetic data}

---

**Algorithm 4** Bagging with resampling

---

**Require:**

Set S containing X input features and Y observations, $(x_1, y_1), ..., (x_m, y_m)$

Learner, $f()$

Number of iterations, $T$

Resampling function, $\mathbf{resample}()$

**for** $t = 1, 2, ... T$ **do**

    $S'_t, D'_t \leftarrow \mathbf{resample}(S_t, D_t)$

    $\mathbf{train} \, f(S'_t, D'_t)$ {train learner using resampled examples}

**end for**

---

---

**Algorithm 5** AdaBoost.RT with resampling

---

**Require:**

Set S containing X input features and Y observations, $(x_1, y_1), ..., (x_m, y_m)$

Learner, $f()$

Number of iterations, $T$

Resampling function, **resample**$()$

Relative error threshold $\phi$

$D_1(i) \leftarrow \frac{1}{m}$ **for** $i = 1, ..., m$ {initialise weights array}

**for** $t = 1, 2, ...T$ **do**

    $S', D'_t \leftarrow$ **resample**$(S, D_t)$

    **train** $f_t(S'_t, D'_t)$ {train learner using resampled examples and weights}

    $\epsilon_t = \sum D_t(i), i = |\frac{(f_t(x_i) - y_i)}{y_i}| > \phi$ {calculate error rate}

    $\beta_t = \epsilon_t^2$

    $D_{t+1}(i) = \frac{D_t(i)}{Z_t} \times \begin{cases} \beta_t, & \text{if } |\frac{(f_t(x_i) - y_i)}{y_i}| \leq \phi \\ 1, & \text{otherwise.} \end{cases}$ {update weights for next boosting iteration}

    $D_{t+1} =$ **normalise**$(D_t)$

**end for**

---

<br>

---

**Algorithm 6** LSBoost with resampling

---

**Require:**

Set S containing X input features and Y observations, $(x_1, y_1), ..., (x_m, y_m)$

Learner, $f()$

Number of iterations, $T$

Resampling function, **resample**$()$

Learning rate $\nu \{0 < \nu \leq 1$

$\hat{Y}_0 = \bar{Y}$

**for** $t = 1, 2, ...T$ **do**

    $R_t = Y - \hat{Y}_{t-1}$

    $S' \leftarrow$ **resample**$(S)$ {resample input features and residuals}

    $R'_t = Y' - \hat{Y}_0 + \sum_{t=1}^{T} \rho_t f_t(X')$ {calculate the residuals corresponding the resampled data}

    **train** $f_t(X', R'_t)$ {train learner to latest residuals}

    $\rho_t = \text{argmin} \sum [\hat{R}_t - \rho R_t]^2$

    $\hat{Y}_t = \hat{Y}_{t-1} + \nu \rho_t f_t(X)$

**end for**

---

*Data availability.* The authors cannot redistribute the data used in this research and must be obtained a request to the respective organisation. The temporal data used in this research may be obtained from the City of Calgary (Bow River precipitation and temperature), the Toronto and Region Conservation Authority (Don River precipitation and stage), Environment Canada (Don River temperature), and the Water Survey of Canada (Bow River stage). Figure 1 was produced using data from the following sources: Esri (aerial basemap (Esri, 2020)), DMTI Spatial

Inc. accessed via Scholars GeoPortal (surface water and Bow River watershed boundary (DMTI Spatial Inc., 2014a, b, c, 2019)), and the TRCA (Don River watershed boundary(Toronto and Region Conservation Authority, 2020b)). Monitoring station locations were obtained from the metadata for the respective temporal datasets.

*Author contributions.* **E. Snieder:** conceptualisation; data curation; formal analysis; investigation; methodology; visualisation; writing - original draft. **K. Abogadil:** review of literature; writing - draft, editing. **U. T. Khan:** conceptualisation; funding acquisition; supervision;

writing - editing, revisions.

*Competing interests.* The authors declare that they have no known competing financial interests or personal relationships that could have appeared to influence the work reported in this paper.

*Disclaimer.* The views expressed in this paper are those of the authors and do not necessarily reflect the views or policies of the affiliated organisations.

*Acknowledgements.* The authors would like to thank the City of Calgary, the Toronto and Region Conservation Authority, and Environment Canada for providing data used in this research.

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
