# Peer review of "Resampling and ensemble techniques for improving ANN-based high flow forecast accuracy"

_Hydrology and Earth System Sciences, 2020_

## Referee Comment (RC1) · Anonymous Referee #1 · 4 Nov 2020

This manuscript compares different resampling methods and different ensemble-building techniques to improve ANN-based flood forecasts (high streamflow). Those resampling methods and ensemble techniques are also combined, resulting in a total of 16 variants, that are compared to a base model. The base model is a classic Multilayer Perceptron with 10 hidden neurons and 25 input variables, trained using a stop training approach and the Levenberg-Marquardt algorithm. All the 16 variants and the base model are applied to simulate (I think) the streamflow for two rivers in Canada.

The manuscript is very well written, well organized and very clear. However, I am sorry to say that I find the originality and contribution to be very low, too low in my opinion

for a publication in HESS. The research was certainly conducted with great care, but ensemble techniques have already been used for quite a while in hydrology (both in the hydro-informatics community and beyond). While the authors mention that most of the resampling and ensemble techniques they used are not common in hydrology, this is still just an additional application of existing techniques, some of which have already been compared. Further, there is very little discussion and analysis of the results. The author conclude that boosting methods provide only marginal improvements, they offer very little explanation.

In addition, I am not convinced that improving only specifically high flows, sometimes at the expense of what the authors call "typical flows", is the way to go. I also fail to see it as a major contribution to the hydrological sciences, worthy of publication in an international journal.

I will still detail some major and minor comments below, but my main concern regarding this paper is the level of the contribution, that I unfortunately find too low.

*Detailed comments (appart from the contribution/originality issue):*

- Abstract, lines 1-2: The affirmation "(…) are increasingly used for operational flood warning systems" should be supported by references to operational flood forecasting systems (not journal articles but documentation from company websites or government websites). At the moment, the affirmation is not only unsupported, but also opposite to my experience. To the best of my knowledge, there are very, very few operational agencies who use ANNs for flood forecasting, despite their use in research for more than 25 years. Also, this affirmation in the abstract somewhat contradicts page 10 lines 157-158 ("Consequently, such models may not be suitable for flood related applications such as flood warning systems").

- Page 2 lines 24-30: One of the causes of errors when simulating high flows in

northern locations such as Canada is the occurence of ice jams. Ice jams are very common and not accounted for in any way by typical hydrological models. Maybe it is possible to account for them using ANNs, but I'm not sure. I think this is one major aspect that should have been present in the manuscript, both on page 2 but also when presenting the Bow River and Don River (it would be important that those rivers be ice jams free, otherwise you have to account for that). Another issue regarding high streamflow that is not discussed by the authors is the fact that those readings are extrapolations from the rating curve. The rating curve of a gauging station is typically constructed with very few (if any) observations of very high streamflow. Therefore, this part of the rating curve is very uncertain, and this is what we use to obtain "observations".

• Page 6, Table 2 and also Page 7 line 128: Why did you use the Levenberg-Marquardt algorithm? Although it is a popular algorithm, it was shown to have oscillation problems around local minima. I think the use of this algorithm should be better justified. See for instance Kwak et al. (2011)

• Page 6 line 116: What is the forecasting horizon? You mention the word "predict" here, but from reading the manuscript it seems like you perform simulations. If they are really forecasts, I think the forecasting horizon should be specified.

• Page 10 line 170: I think there is a mistake here "(. . .) studies that featuring each (. . .)"

• Page 11: the distinction between RUS and ROS seems extremely thin to me.

• Page 13 lines 256-257: The definition of ESP that you provide here corresponds to *Extended* Streamflow Prediction, as per Day (1985), not *Ensemble* stream-flow predictions. Ensemble streamflow forecasts (or predictions) can be obtained by a variety of manners, including feeding a hydrological model with dynamical meteorological ensemble forecasts.

- Page 14: Why 20 members?

- Results: Why do you aggregate the ensemble into a deterministic simulation and therefore evacuate the information about the uncertainty? Why not use ensemble-based performance assessment criterion such as the Continuous Ranked Probability Score, logarithmic score, etc.?

- Page 19 lines 457-459: How can overtraining happen if you have used stop training? I think this has to be explained more.

- Page 21, Figure 7: From a general perspective, I don't see how decreasing the quality of simulation for typical flow values could be positive, even if the simulation of high flows is improved. Typical and especially very low flows are also important.

- Page 26 lines 531-536: The analysis and discussion are very thin.

- Page 26, section 4.2: I disagree with the idea of fine tuning the hyper parameters differently and specifically for each model, as it would violate the ceteris paribus principle, making it difficult to isolate and compare the influence of ensemble techniques and resampling techniques.

- Page 26-27 lines 562-563: Ensembles are already quite common in ANN-based flow forecasting model, so this is not a very useful recommandation.

References:

Day, G. (1985). "Extended Streamflow Forecasting Using NWSRFS." J. Wat. Res. Plan. Mgmt., 10.1061/(ASCE)0733-9496(1985)111:2(157), 157-170

Kwak Y., Hwang J. and Yoo C-J. (2011). A new damping strategy of Levenberg-Marquardt algorithm for Multilayer Perceptrons, Neural Network World, 21, 327-340

---

## Referee Comment (RC2) · Anonymous Referee #2 · 6 Nov 2020

The authors study the effect of resampling techniques, when integrated with ensemble learning frameworks, on the ability of the ANN based regression ensemble learners to improve prediction of high steam flow events. Two case studies are presented, with different temporal resolution and, essentially, hydrologic topology. One individual learner, that is MLP-ANN, is utilized in this study along with two ensemble models (Bagging and Boosting) as well as a randomized set of members (i.e. RWB model). Three resampling plans are examined, RUS, ROS, and SMOTER, to serve as the preprocessing re-sampler stage for the ensemble models. a combination of the latter is used with the ensemble models and all configurations are evaluated.

This paper attempts to answer an important question which is usually overlooked; can we diminish the heteroscedastic nature of stream flow predictions, which is inevitable when dealing with limited intelligence about the system dynamics (drivers to the instantaneous change in streamflow). The authors are concerned with the most volatile aspect in this setting, high flows, and whether more information can be utilized from the available descriptors' database to alleviate the problem. In general, ensemble learning is one of the few state-of-the-art solutions for improved short- to mid-term streamflow prediction, as individual nonlinear models are inherently instable and conventional statistical approaches sacrifices accuracy for probabilistic interpretability. As the diversity-in-learning mechanism, promoted differently by each ensemble architecture, is assumed to be the major reason to ensemble's generalization ability, resampling techniques are a major interest here. Consequently, it makes sense to study variations of ensemble learning frameworks with respect to the utilized resampling approach. This topic is increasingly gaining attention in the recent, and highly evolving, applied ML community in the field.

The paper is well-written (the chronologic format and section types are not similar to that I am used to, though). The presented results are critical, valid, and to-the-point. The resampling methods are described in organized wording and supported with references. The discussion covers most of the important aspects of this research. To this extent, I believe this paper meets HESS standards and scope, and is worthy of publication, after few important additions and modifications are implemented. Please refer below to the major and minor comments for consideration by the authors.

1. The summarization of the individual model calibration is good enough, as the focus is on the resampling-ensemble models. The use of the PC operator to select features from the predetermined bag of lags also makes sense. However, the authors should track the isolated features and report them. Are they used uniformly among all ensembles? Or does the PC based selection changes per ensemble? Could you also elaborate of the significance of the selected features as well (important for semi-physical

validation of the used features).

2. Please include pseudo-algorithm table for each resampling plan. This is very important for recreation proposes (the two utilized ensemble models, on the other hand, are well-studied in the broad literature and do not require detailed description; though I would prefer to see a mathematical description of the models to further acquaint the readers with them as ensemble learning is still not common across all fields of HESS).

3. Also, please elaborate more on the distinction between RUS and ROS (for a first glance, the wording makes RUS looks like a special case of ROS, but they are very different in reality and have distinct effect on the model performance). Please elaborate more on your choice of the ROS configuration, and why not present an array of results related to the OSR (ratio vs. performance for example).

4. Please modify the ensemble learning section to have a more concise summary of ensemble learning, diversity-in-learning concept and the effect of resampling as part of the latter. Please provide references from the pure literature.

5. It is important to note that the RWB model, contrast to what has been suggested in the applied literature, is NOT an ensemble model. Ensemble learning has three stages one of which is the resampling of the available intelligence. RWB violates this and should not be considered an ensemble for the sake of clarity. However, the randomization of weights within the individual learners are a major source of diversity, as shown in the literature, which promotes similar behaviour improvement in this model as other ensembles. RWB can be considered as middle ground between individual and ensemble learning. Please fix this issue.

6. I do not think that the authors replace the ensemble's native resampling technique with the suggested approaches, but rather add them as a preprocessing phase of the available data, as indicated in the manuscript (also, it would be impossible to replace the resampling approach in boosting!). Hence, it seems like there is "double resampling" which occurs in the modified ensemble models. this is interesting (as shown in

the obtained results). Please elaborate on the underlying effect of the ensemble's native resampling technique on the preprocessing one. For example, Bagging's uniform resampling plan has little effect, while that of Boosting has a very strong effect (I am thinking it nullifies the preprocessing resampler!).

7. Linking the previous comment with RWB, having a preprocessing resampler makes RWB a true ensemble model here. More importantly, the double-resampling effect is absent here, making it a great opportunity to elaborate on the difference between modified RWB and other ensembles! Please do so.

8. Figures 6 to 9 and Tables 5 and 6 are very important and provide most of the critical information to show how added resampling promotes improved high-flow prediction (and overall prediction also). In the same time, I sincerely think that a major result is missing here, which deals with quantifying the effect of the resampling approaches on the ensembles. I think that the paper requires a figure showing the change of ensemble performance versus ensemble size. This figure should at least depict the Bagging model and the modified Bagging model. I recommend adding the RWB and Modified RWB. This is very important to cross-examine the change of performance, per ensemble size, between the normal and modified ensembles and provide more insight on the effect of the preprocessing phase.

9. The fact that tables 5 and 6 show Boosting to perform the worst validates the results obtained, as Boosting performance deteriorates in the presence od "hard instances" in general and is more applicable to classification applications (at a larger ensemble size, the obtained combiners' weights seem to dilute in efficacy when performing regression. But in classification, they are powerful especially in binary classification due to the sign significance rather than magnitude of the weight). The authors attempt to explain the reasons to the deficiency of Boosting ensemble in the paper but I think they can elaborate more.

10. More importantly, the information in comments 3, 6 and 7 should be considered

here (when discussing tables 5 and 6). For example in table 5, when considering RWB and Bagging models, why was ROS based models the worst (I think because there are a lot of data), but in the same time SMOTER variations did provide good performance in few of the metrics. Why did SMOTED-RWB have an unusually low PITF but good performance with respect to other metrics? Please provide more details similar to this.

11. There are few typos and referencing issues. Please revise the manuscript for this. A few examples are provided below:

o Line 4: "compare three resampling;" I think it is missing a word! o Table 3: second column. Do you mean "variable" or "feature"? o Line 113: please include the term "individual learners". o Line 163: comma is missing after the reference. o Line 164: the reference seems to be in the middle of the sentence. I noticed you do this often. Please try to minimize this to enable smother flow of the idea. o Line 168: "for handling imbalance data..." do you mean "imbalanced". o Line 170: "that featuring each...". Please fix. o Please italicize all symbols or feature names, such as $N$, $theta$, $PI$, $CE$, etc., across the manuscript. o Line 205: the reference format needs a fix. o Line 255 to 260: please re-write the ensemble learning summary as discussed in the major comment. o Line 262: "Ensembles members are.." please remove the "s". o Line 310: do you mean "regional flood quantiles"?.

---

## Referee Comment (RC3) · K.S. Kasiviswanathan (Referee) · 20 Nov 2020

General: This paper explored the potential of data-driven models such as ANN for improving the accuracy of high flow estimation through integrating resampling and ensemble techniques. For this exercise, three resampling techniques: random undersampling (RUS), random oversampling (ROS), and SMOTER; and four ensemble techniques: randomized weights and biases, bagging, adaptive boosting (AdaBoost), least-squares boosting (LSBoost) were systematically combined to show the improvement in the forecast accuracy in terms of reducing the timing and amplitude error. This paper used the hourly river stage data along with other meteorological data collected

from Bow and Don River basins, Canada to demonstrate the proposed modelling approaches. While many previous papers have already reported the potential application of several ensembles and resampling methods to improve the forecast accuracy of data-driven models, this paper claims that the implementation of ROS, and new approaches for SMOTER, LSBoost, and SMOTER-AdaBoost are the new addition. The paper is well written and interesting to the researchers of hydrology. However, the paper needs some more clarity, which I have marked below. 1. Since the variation in the streamflow is evident, I do not know the usage of word imbalance is correct or not in this context. 2. You have selected the 80th percentile to segregate the peak flow data from the entire dataset. I agree that ANN models are completely dependent on the choice of data. Still, it would be interesting to see the effect of selecting any other values (70th and 90th percentile), at least for a few cases. 3. How to choose the model HF, TF for the unknown data for the future forecast? 4. How do you define highly imbalanced flow datasets? 5. Line 25: "One cause of low model accuracy on high flows is the scarcity of representative sample observations available with which to train such models." Add one or two references 6. Line 30: "As a result, studies that assess models using traditional performance metrics risk overlooking deficiencies in high flow performance." I agree with this point. However, separating high flow hydrograph from the dataset and evaluate the model performance using the traditional indices would still reveal the actual model performance. This should be mentioned. 7. Line 40: Improving the accuracy of high flow forecasts has been the focus of many studies. Several studies have examined the use of preprocessing techniques to improve model performance. Reference is required. I would suggest adding Kasiviswanathan et al. (2015). 8. Line 85: The Bow and Don River watersheds are the focus of this research. You may consider deleting this line. 9. Authors refer to the stage as flow. This should be corrected. 10. It would be interesting to see how the peak flow of Bow River in the years 2005 and 2013 forecasted by these models. Similarly, for Don River. 11. Why stage data, why not directly for the discharge data?

Reference Kasiviswanathan KS, He J, Sudheer KP, Tay J-H (2016) Potential application of wavelet neural network ensemble to forecast streamflow for flood management. Journal of Hydrology, DOI: 10.1016/j.jhydrol.2016.02.04

---

## Author Comment (AC1) · 1 Jan 2021

AC3-General. This paper explored the potential of data-driven models such as ANN for improving the accuracy of high flow estimation through integrating resampling and ensemble techniques. For this exercise, three resampling techniques: random undersampling (RUS), random oversampling (ROS), and SMOTER; and four ensemble techniques: randomized weights and biases, bagging, adaptive boosting (AdaBoost), least-squares boosting (LSBoost) were systematically combined to show the improvement in the forecast accuracy in terms of reducing the timing and amplitude error. This paper used the hourly river stage data along with other meteorological data collected

from Bow and Don River basins, Canada to demonstrate the proposed modelling approaches. While many previous papers have already reported the potential application of several ensembles and resampling methods to improve the forecast accuracy of data-driven models, this paper claims that the implementation of ROS, and new approaches for SMOTER, LSBoost, and SMOTER-AdaBoost are the new addition. The paper is well written and interesting to the researchers of hydrology. However, the paper needs some more clarity, which I have marked below.

RC3-General. Thank you for the positive comments on our manuscript. We have addressed the comments for better clarity below.

RC3-1. Since the variation in the streamflow is evident, I do not know the usage of word imbalance is correct or not in this context.

AC3-1. The term 'imbalance' is widely used in machine learning literature to describe imbalance of labels for classification problems [1]. Many studies have extended the use of this term to regression problems [2]. In simple terms, "imbalanced" is defined as the "existence of an over-representation of a given class(es) or numeric value interval(s), over another" [3].

AC3-2. You have selected the 80th percentile to segregate the peak flow data from the entire dataset. I agree that ANN models are completely dependent on the choice of data. Still, it would be interesting to see the effect of selecting any other values (70th and 90th percentile), at least for a few cases.

RC 3-2. Thank you for this recommendation. A formal grid-search and subsequent discussion on HF thresholds (ranging from 50 to 90th percentile flow) has been added to the revised manuscript (please see Fig 11 and associated text the attached supplement). In this new analysis, we compare the Bagging ensemble with SMOTER-Bagging in order to quantify the relative effects of resampling for various high flow thresholds.

RC3-3. How to choose the model HF, TF for the unknown data for the future forecast?

AC3-3. It is not necessary to know the HF or TF of the future, unknown data - we assume these values to be constant for the simulations. With sufficient historic data, the temporal variance of the high flow threshold is assumed to be negligible. This can be confirmed using statistical bootstrapping: bootstrapping the 80th percentile flow (n = 10,000) returns a standard deviation of 0.0046 and 0.0031 for the Bow and Don River, respectively. Alternatively, as recommended in the text, a high flow threshold could be chosen based on physical characteristics (such as the stage at which a river exceeds its banks).

RC3-4. How do you define highly imbalanced flow datasets?

AC3-4. "Highly imbalanced" is commonly used in literature [4-7]; however, the authors are not aware of a quantitative definition. For that reason, the term "highly" has been removed from the text and instead we simply state "imbalanced".

RC3-5. Line 25: "One cause of low model accuracy on high flows is the scarcity of representative sample observations available with which to train such models." Add one or two references

AC3-5. We appreciate this recommendation and have added the following reference [8].

RC3-6. Line 30: "As a result, studies that assess models using traditional performance metrics risk overlooking deficiencies in high flow performance." I agree with this point. However, separating high flow hydrograph from the dataset and evaluate the model performance using the traditional indices would still reveal the actual model performance. This should be mentioned.

AC3-6. We appreciate this insightful suggestion. In fact, this is exactly what we do in our manuscript; a fixed threshold is used to separate high and typical stage values. This simple approach was favoured over more complex methods of high flow performance assessment such as for hydrograph extraction, which isolate separate hydrological events and baseflow. Large hydrological events during low baseflow periods, which occur in highly seasonal watersheds such as the Bow, are not necessarily relevant to flood forecasting, as the event peak stage may not attain flood-level stage. Comparatively, a small event during high baseflow may reach a higher peak stage. Thus, we found that simply separating high and typical stages based on a fixed threshold is more appropriate than more complex hydrograph extraction methods for applications such as high stage forecasting. We will include a discussion on this in the updated manuscript. In our previous research [9, 10] on improving high flow performance, we demonstrate hydrograph extraction and the use of more complex, event-specific performance measures; however, this approach was abandoned for the aforementioned reasons.

RC3-7. Line 40: Improving the accuracy of high flow forecasts has been the focus of many studies. Several studies have examined the use of preprocessing techniques to improve model performance. Reference is required. I would suggest adding Kasiviswanathan et al. (2015).

AC3-7. Thank you for this suggestion, we have added this citation in the manuscript. Preprocessing methods used in several works cited in the original draft: [11] evaluates statistical transformations and [12] evaluates a multi-model approach based on cluster-based preprocessing that evaluate preprocessing techniques for improving high flow performance.

RC3-8. Line 85: The Bow and Don River watersheds are the focus of this research. You may consider deleting this line.

AC3-8. Thank you for this recommendation. The focus of the research is indeed the ensemble and resampling methods. The sentence has been modified as follows: "The Bow and Don Rivers are featured as case studies in this research to evaluate methods for improving the accuracy of high stage data-driven forecasts"

RC3-9. Authors refer to the stage as flow. This should be corrected.

AC3-9. Thank you for this recommendation. In our original submission, we refer to flow forecasting but opt to use stage data in our models, as stage in forecasts are more useful than flow for indicating flooding and stage-discharge curves are readily available for both rivers. We understand how this may cause confusion and have changed all uses of 'flow' to 'stage', where appropriate.

RC3-10. It would be interesting to see how the peak flow of Bow River in the years 2005 and 2013 forecasted by these models. Similarly, for Don River.

AC3-10. The results for the Bow River for 2005 were included in the original manuscript (Fig. 3). Individual performance for this calendar year is reported in Table RC3-1 (please refer to attached supplement).

We currently do not have access to the 2013 Bow River data but will consider adding it to the analysis for the final revised paper. For the Don River, the high resolution data needed for our research is unavailable for 2005 and 2013 for some of the hydrometeo-rological stations used in this research.

However, note that the current dataset is highly imbalanced as shown in Fig. 2 in the original manuscript. Thus, we think the data we used is sufficient for the analysis to demonstrate the effects of resampling/ensemble with respect to the data imbalance problem, even without the additional data for 2005 and 2013 for each river.

RC3-11. Why stage data, why not directly for the discharge data?

AC3-11. Stage data is used for several reasons. Foremost, it is more relevant, compared to discharge, for an early flood warning system, which is the anticipated application of this research. Next, stage is measured directly, whereas flow is calculated based on an uncertain stage-discharge relationship. References to flow or discharge have been corrected to flow, following RC3-9.

References [1] M. Galar, A. Fernandez, E. Barrenechea, H. Bustince, and F. Herrera, "A review on ensembles for the class imbalance problem: Bagging-, boosting-, and

hybrid-based approaches," IEEE Transactions on Systems, Man and Cybernetics Part C: Applications and Reviews, vol. 42, no. 4. pp. 463–484, 2012.

[2] P. Branco, L. Torgo, and R. P. Ribeiro, "A survey of predictive modeling on imbalanced domains," ACM Computing Surveys, vol. 49, no. 2. Association for Computing Machinery, p. 31, 01-Aug-2016.

[3] N. Moniz, P. Branco, L. Torgo, and B. Krawczyk, "Evaluation of Ensemble Methods in Imbalanced Regression Tasks," Proc. Mach. Learn. Res., vol. 74, pp. 129–140, 2017.

[4] M. Galar, A. Fernández, E. Barrenechea, and F. Herrera, "EUSBoost: Enhancing ensembles for highly imbalanced data-sets by evolutionary undersampling," Pattern Recognit., vol. 46, no. 12, pp. 3460–3471, Dec. 2013.

[5] G. Haixiang, L. Yijing, J. Shang, G. Mingyun, H. Yuanyue, and G. Bing, "Learning from class-imbalanced data: Review of methods and applications," Expert Systems with Applications, vol. 73. Elsevier Ltd, pp. 220–239, 01-May-2017.

[6] P. Branco, L. Torgo, and R. P. Ribeiro, "A survey of predictive modeling on imbalanced domains," ACM Computing Surveys, vol. 49, no. 2. Association for Computing Machinery, p. 31, 01-Aug-2016.

[7] O. Loyola-González, J. F. Martínez-Trinidad, J. A. Carrasco-Ochoa, and M. García-Borroto, "Study of the impact of resampling methods for contrast pattern based classifiers in imbalanced databases," Neurocomputing, vol. 175, pp. 935–947, 2016.

[8] N. Moniz, P. Branco, and L. Torgo, "Resampling strategies for imbalanced time series forecasting," Int. J. Data Sci. Anal., vol. 3, no. 3, pp. 161–181, May 2017.

[9] E. J. Snieder, "Artificial Neural Network-Based Flood Forecasting: Input Variable Selection and Peak Flow Prediction Accuracy," Master's Thesis, York University, 2019. Available at: http://hdl.handle.net/10315/36792 [10] U. T. Khan and E. Snieder, "Assessment of peak flow error metrics and error weights for data-driven riverine flow fore-

casting models.," in Geophysical Research Abstracts, Proceedings of the European Geosciences Union General Assembly, 2019, vol. 21.

[11] K. P. Sudheer, P. C. Nayak, and K. S. Ramasastri, "Improving peak flow estimates in artificial neural network river flow models," Hydrol. Process., vol. 17, no. 3, pp. 677–686, Feb. 2003.

[12] W. Wang, P. H. A. J. M. V. Gelder, J. K. Vrijling, and J. Ma, "Forecasting daily streamflow using hybrid ANN models," J. Hydrol., vol. 324, no. 1–4, pp. 383–399, 2006.

Please also note the supplement to this comment:
https://hess.copernicus.org/preprints/hess-2020-430/hess-2020-430-AC1-supplement.pdf

[Figure]

**Supplement:**

[Figure]

Figure 11. Calibration and test MSE ratio between Bagging and SMOTER-Bagging models for the Bow (a) and (c) and Don (b) and (d) Rivers across high stage threshold values ranging from 50% to 90%.

"As discussed in Sect. 2.1, a fixed threshold is used to distinguish between high and typical stages. Fig. 11 shows the effects of the fixed threshold increasing from the 50th to 90th percentile of the stage distribution. These plots show the relative effects of SMOTER-Bagging compared to simple Bagging. A performance ratio greater than 1 indicates that the SMOTER-Bagging model has greater error compared to the Bagging model, 1 indicates that they have the same performance, and less than 1, improved performance. The error (MSE) is presented for all stages as well as the TS and HS subsets. The calibration plots illustrate an asymmetric trade-off between HS and TS error. For a given $\theta_{HS}$ value, the error ratio of the TS subset increases more than the decline in HS error. More importantly, the improvements in HS performance obtained in calibration are considerably less pronounced in the test dataset, despite a loss in TS performance".

Table RC3-1: Bow River 2005 performance for the testing fold

| Label | ce_cal | ce_tf_cal | ce_hf_cal | pi_cal | pi_tf_cal | pi_hf_cal | ce_test | ce_tf_test | ce_hf_test | pi_test | pi_tf_test | pi_hf_test |
|---|---|---|---|---|---|---|---|---|---|---|---|---|
| Single | 0.98 | 0.96 | 0.89 | 0.23 | 0.18 | 0.28 | 0.93 | 0.95 | 0.69 | 0.14 | 0.25 | 0.11 |
| RWB | 0.98 | 0.96 | 0.91 | 0.30 | 0.22 | 0.39 | 0.93 | 0.96 | 0.67 | 0.12 | 0.30 | 0.07 |
| RUS-RWB | 0.98 | 0.96 | 0.91 | 0.27 | 0.15 | 0.40 | 0.93 | 0.95 | 0.66 | 0.09 | 0.27 | 0.05 |
| ROS-RWB | 0.98 | 0.96 | 0.91 | 0.28 | 0.16 | 0.42 | 0.92 | 0.95 | 0.64 | 0.03 | 0.26 | -0.03 |
| SMOTER-RWB | 0.98 | 0.96 | 0.92 | 0.30 | 0.17 | 0.44 | 0.93 | 0.96 | 0.68 | 0.13 | 0.29 | 0.09 |
| Bagging | 0.98 | 0.96 | 0.91 | 0.29 | 0.22 | 0.37 | 0.92 | 0.96 | 0.62 | -0.01 | 0.31 | -0.09 |
| RUS-Bagging | 0.98 | 0.96 | 0.91 | 0.28 | 0.16 | 0.41 | 0.92 | 0.96 | 0.64 | 0.04 | 0.30 | -0.03 |
| ROS-Bagging | 0.98 | 0.96 | 0.92 | 0.29 | 0.15 | 0.43 | 0.92 | 0.95 | 0.61 | -0.04 | 0.27 | -0.11 |
| SMOTER-Bagging | 0.98 | 0.96 | 0.92 | 0.28 | 0.16 | 0.42 | 0.93 | 0.96 | 0.65 | 0.07 | 0.29 | 0.02 |
| AdaBoost | 0.98 | 0.96 | 0.91 | 0.29 | 0.20 | 0.38 | 0.92 | 0.96 | 0.64 | 0.04 | 0.31 | -0.02 |
| RUS-AdaBoost | 0.98 | 0.96 | 0.91 | 0.27 | 0.14 | 0.39 | 0.91 | 0.95 | 0.58 | -0.08 | 0.28 | -0.17 |
| ROS-AdaBoost | 0.98 | 0.95 | 0.91 | 0.25 | 0.10 | 0.41 | 0.93 | 0.95 | 0.66 | 0.08 | 0.26 | 0.03 |
| SMOTER-AdaBoost | 0.98 | 0.96 | 0.92 | 0.29 | 0.17 | 0.43 | 0.92 | 0.96 | 0.61 | -0.01 | 0.31 | -0.09 |
| LSBoost | 0.98 | 0.96 | 0.91 | 0.29 | 0.19 | 0.38 | 0.93 | 0.96 | 0.69 | 0.15 | 0.30 | 0.11 |
| RUS-LSBoost | 0.98 | 0.95 | 0.92 | 0.28 | 0.09 | 0.48 | 0.92 | 0.94 | 0.64 | 0.01 | 0.12 | -0.02 |
| ROS-LSBoost | 0.98 | 0.95 | 0.92 | 0.26 | 0.07 | 0.45 | 0.86 | 0.95 | 0.32 | -0.70 | 0.24 | -0.93 |
| SMOTER-LSBoost | 0.98 | 0.95 | 0.91 | 0.25 | 0.10 | 0.40 | 0.93 | 0.96 | 0.67 | 0.12 | 0.29 | 0.07 |

---

## Author Comment (AC2) · 1 Jan 2021

RC2-General.The authors study the effect of resampling techniques, when integrated with ensemble learning frameworks, on the ability of the ANN based regression ensemble learners to improve prediction of high steam flow events. Two case studies are presented, with different temporal resolution and, essentially, hydrologic topology. One individual learner, that is MLP-ANN, is utilized in this study along with two ensemble models (Bagging and Boosting) as well as a randomized set of members (i.e. RWB model). Three resampling plans are examined, RUS, ROS, and SMOTER, to serve as the preprocessing re-sampler stage for the ensemble models. a combination of the

latter is used with the ensemble models and all configurations are evaluated.

This paper attempts to answer an important question which is usually overlooked; can we diminish the heteroscedastic nature of stream flow predictions, which is inevitable when dealing with limited intelligence about the system dynamics (drivers to the instantaneous change in streamflow). The authors are concerned with the most volatile aspect in this setting, high flows, and whether more information can be utilized from the available descriptors' database to alleviate the problem. In general, ensemble learning is one of the few state-of-the-art solutions for improved short- to mid-term streamflow prediction, as individual nonlinear models are inherently instable and conventional statistical approaches sacrifices accuracy for probabilistic interpretability. As the diversity-in-learning mechanism, promoted differently by each ensemble architecture, is assumed to be the major reason to ensemble's generalization ability, resampling techniques are a major interest here. Consequently, it makes sense to study variations of ensemble learning frameworks with respect to the utilized resampling approach. This topic is increasingly gaining attention in the recent, and highly evolving, applied ML community in the field.

The paper is well-written (the chronologic format and section types are not similar to that I am used to, though). The presented results are critical, valid, and to-the-point. The resampling methods are described in organized wording and supported with references. The discussion covers most of the important aspects of this research. To this extent, I believe this paper meets HESS standards and scope, and is worthy of publication, after few important additions and modifications are implemented. Please refer below to the major and minor comments for consideration by the authors.

AC2-General. Thank you for the positive comments on our manuscript and for highlighting the need for the present research. We have addressed each of the major and minor comments below.

RC2-1. The summarization of the individual model calibration is good enough, as the

focus is on the resampling-ensemble models. The use of the PC operator to select features from the predetermined bag of lags also makes sense. However, the authors should track the isolated features and report them. Are they used uniformly among all ensembles? Or does the PC based selection changes per ensemble? Could you also elaborate of the significance of the selected features as well (important for semi-physical validation of the used features).

AC2-1. Thank you for the recommendation. The selected input features are now listed in Table A1 of the Appendix (please refer to attached supplement). Since PC is a model free method, the outcome is independent from the training method, thus the feature set is constant for each method. In fact, this is the reason we selected to use the PC method, since it allows a consistent feature set for all model configurations. As for the semi-physical validation of the features, this is discussed in detail in [1]. However, in general, due to the autocorrelated nature of the Bow River, the inputs are dominated by autoregressive input variables, upstream flow, and temperature, which drives snowmelt. Whereas, for the Don River, the inputs are a mixture of precipitation and upstream and/or lagged flows. This to be expected, based on results in [1], as well as the nature of the two watersheds.

RC2-2. Please include pseudo-algorithm table for each resampling plan. This is very important for recreation proposes (the two utilized ensemble models, on the other hand, are well-studied in the broad literature and do not require detailed description; though I would prefer to see a mathematical description of the models to further acquaint the readers with them as ensemble learning is still not common across all fields of HESS).

AC2-2. Thank you for the recommendation. Pseudocode has been provided for the three resampling methods and three ensemble methods (please refer to the attached supplement). Including the code for the ensemble methods (in addition to the resampling plans) is also relevant, as the resampling methods are embedded within ensemble methods.Thus, both have been included.

RC2-3. Also, please elaborate more on the distinction between RUS and ROS (for a first glance, the wording makes RUS looks like a special case of ROS, but they are very different in reality and have distinct effect on the model performance). Please elaborate more on your choice of the ROS configuration, and why not present an array of results related to the OSR (ratio vs. performance for example).

You are correct: ROS and RUS are distinct (i.e., RUS is not a special case of ROS). As stated in-text, RUS undersamples data as to achieve a balanced training set with no duplicates whereas ROS includes all available data and creates duplicates. Some studies have compared the two sampling techniques, however neither one of these two methods consistently outperforms the other, thus both are included in our study [2-4]. We have now included the pseudo-algorithms for each (see response AC2-2 above) to help clarify the distinction. All three resampling methods (RUS, ROS, and SMOTER) are configured such that the number of typical and high stage samples in the resampled dataset are equal. In this research, since the 80th percentile stage is used to distinguish between typical and high flows, ROS resamples the high stage subset by 500%.

Furthermore, the effects of SMOTER-based resampling have been explicitly quantified by calculating the ratio between SMOTER-Bagging and Bagging in two new figures. Firstly, the effects of SMOTER resampling is assessed across the high stage threshold value, thus rate of oversampling. Next, the resampling effects are analysed across the number of ensemble members, thus the number of resampling repetitions. Please refer to Figures 11 and 12, and associated text in the attached supplement.

RC2-4. Please modify the ensemble learning section to have a more concise summary of ensemble learning, diversity-in-learning concept and the effect of resampling as part of the latter. Please provide references from the pure literature.

AC2-4. We apologise for the verbose section on ensemble learning and thank you for this recommendation. Some of the oversimple background on ensemble methods have

been trimmed from the text. Relevant background on ensemble learning and sources of diversity have been added to the text and collection of cited works have been expanded: "Ensembles are collections of models, each trained on different subsets of the available training data and combined to form discrete ensemble prediction (Alobaidi et al., 2019). It is well established that ensemble-based methods improve model stability and generalisability (Alobaidi et al., 2019; Brown et al., 2005). Recent advances in ensemble learning have emphasised the importance of diversity-in-learning (Alobaidi et al., 2019). Diversity can be generated both implicitly and explicitly through a variety of methods, some of which include varying the initial set of model parameters, varying the model topology, varying the training algorithm, and varying the training data (Sharkey, 1996; Brown et al., 2005). The largest source of diversity in the ensembles under study is attributable with varying the training data, which occurs both in the various resampling methods described above and the in some cases, the ensemble algorithms. Only homogeneous ensembles are used in this work, thus no diversity is obtained through varying the model topology or training algorithm (Zhang et al., 2018; Alobaidi et al., 2019). Ensemble predictions are combined to form a single discrete prediction. Ensembles that are combined to produce discrete predictions have been proven to outperform single models by reducing model bias and variance, thus improving overall model generalisability (Brown et al., 2005; Sharkey, 1996; Shu and Burn, 2004; Alobaidi et al., 2019). This has contributed to their widespread application in hydrological modelling (Abrahart et al., 2012). In some cases, ensembles are not combined, and the collection of predictions are used to estimate the uncertainty associated with the diversity between ensemble members (Tiwari and Chatterjee, 2010; Abrahart et al., 2012). While this approach has obvious advantages, it is not possible for all types of ensembles, such as the boosting methods used in this work."

RC2-5. It is important to note that the RWB model, contrast to what has been suggested in the applied literature, is NOT an ensemble model. Ensemble learning has three stages one of which is the resampling of the available intelligence. RWB violates this and should not be considered an ensemble for the sake of clarity. However, the

randomization of weights within the individual learners are a major source of diversity, as shown in the literature, which promotes similar behaviour improvement in this model as other ensembles. RWB can be considered as middle ground between individual and ensemble learning. Please fix this issue.

AC2-5. Thank you for clarifying this point. While we are not aware of an agreed-upon definition for what qualifies an ensemble model in literature [5,6]. Quoting from [6]: "This is perhaps the most common way of generating an ensemble, but is now generally accepted as the least effective method of achieving good diversity; many authors use this as a default benchmark for their own methods" [6].

You are correct that RWB is unlike the other ensemble methods in that it does not have a source of diversity-in-learning. We have kept the organisation of the manuscript the same, but added text to distinguish RWB from the ensemble learning methods in this regard. We have adjusted the RWB text accordingly: "While not technically a form of ensemble learning, repeatedly randomising the weights and biases of ANNs is one of the simplest and most common methods for achieving diversity among a collection of models, thus it acts as a good comparison point for the proceeding ensemble methods (Brown et al., 2005). In this method, members are only distinguished by the randomisation of the initial parameter values (i.e., the initial weights and biases for ANNs in this research) used for training."

RC2-6. I do not think that the authors replace the ensemble's native resampling technique with the suggested approaches, but rather add them as a preprocessing phase of the available data, as indicated in the manuscript (also, it would be impossible to replace the resampling approach in boosting!). Hence, it seems like there is "double resampling" which occurs in the modified ensemble models. this is interesting (as shown in the obtained results). Please elaborate on the underlying effect of the ensemble's native resampling technique on the preprocessing one. For example, Bagging's uniform resampling plan has little effect, while that of Boosting has a very strong effect (I am thinking it nullifies the preprocessing resampler!).

AC2-6. Thank you for the recommendation. We do, in fact, imbed the resampling methods within the ensemble algorithms. In machine learning literature this has been referred to as hybridisation [7]. For example, the hybridisation of oversampling with bagging achieves the benefits of increased representation of infrequent values (high flows) and the diversity obtained through repeated resampling with replacement [7]. The most complex of these combinations, AdaBoost with SMOTE, has been demonstrated for classification [8] and regression [9] in existing literature. AdaBoost is configured with reweighting the ANN cost function, so that there is no double resampling; however, such a resampling scheme is possible. We believe that our proposed method, for determining sample weights for synthetic samples in this algorithm, is novel and an improvement over existing implementations of SMOTE with AdaBoost. The pseudocode produced in response to RC2-2 elaborates on the implementation of these methods.

RC2-7. Linking the previous comment with RWB, having a preprocessing resampler makes RWB a true ensemble model here. More importantly, the double-resampling effect is absent here, making it a great opportunity to elaborate on the difference between modified RWB and other ensembles! Please do so.

AC2-7. As stated in our response to RC2-6, the resampling methods are embedded within the ensemble methods. Subsequently, for RWB, resampling occurs as a one-time preprocessing step and the series of models with randomised weights and biases are trained using the fixed, resampled data. The ensemble methods do not have double resampling, as the resampling methods are not used in preprocessing. Please refer to the pseudocodes above (AC2-2) for a description of the methods.

RC2-8. Figures 6 to 9 and Tables 5 and 6 are very important and provide most of the critical information to show how added resampling promotes improved high-flow prediction (and overall prediction also). In the same time, I sincerely think that a major result is missing here, which deals with quantifying the effect of the resampling approaches on the ensembles. I think that the paper requires a figure showing the

change of ensemble performance versus ensemble size. This figure should at least depict the Bagging model and the modified Bagging model. I recommend adding the RWB and Modified RWB. This is very important to cross-examine the change of performance, per ensemble size, between the normal and modified ensembles and provide more insight on the effect of the preprocessing phase.

AC2-8. Thank you for this recommendation. We have included a formalised ensemble size grid-search for ensemble sizes ranging from 2-100 for each of the base ensemble methods (i.e., with no added resampling). Please refer to Fig. 6 and associated text in the attached supplement.

We have also conducted a grid-search of ensemble size to assess the SMOTER resampling effect on the Bagging algorithm. The new figure (Fig. 12) and associated text is included in the attached supplement.

RC2-9. The fact that tables 5 and 6 show Boosting to perform the worst validates the results obtained, as Boosting performance deteriorates in the presence of "hard instances" in general and is more applicable to classification applications (at a larger ensemble size, the obtained combiners' weights seem to dilute in efficacy when performing regression. But in classification, they are powerful especially in binary classification due to the sign significance rather than magnitude of the weight). The authors attempt to explain the reasons to the deficiency of Boosting ensemble in the paper but I think they can elaborate more.

AC2-9. The authors have added text addressing the overfitting tendencies of the boosted ensembles:

"The overfitting produced by the boosting methods is consistent with previous research, which finds that boosting is sometimes prone to overfitting on real-world datasets (Vezhnevets and Barinova, 2007). One reason that the improvements made by the boosting methods (AdaBoost and LSBoost) are not more substantial may be due to the use of ANNs as individual learners. ANNs typically have more degrees of freedom

compared to the decision trees that are most commonly used as individual learners; thus, the additional complexity offered by boosting does little to improve model predictions. Additionally, the boosting methods further increase the effective degrees of freedom of the predictions. Nevertheless, these methods still tend to improve performance over the base model case. Ensembles of less complex models such as regression trees are expected to produce relatively larger improvements when relative to the single model predictions."

RC2-10. More importantly, the information in comments 3, 6 and 7 should be considered here (when discussing tables 5 and 6). For example in table 5, when considering RWB and Bagging models, why was ROS based models the worst (I think because there are a lot of data), but in the same time SMOTER variations did provide good performance in few of the metrics. Why did SMOTED-RWB have an unusually low PITF but good performance with respect to other metrics? Please provide more details similar to this.

AC2-10. Thank you for this insightful recommendation. We have added specific remarks on the overfitting effects of ROS and the difference in performance and diversity between RUS-RWB and RUS-Bagging. The text is copied below, and this has also been addressed elsewhere in the response to the reviewer's comments:

"ROS often exhibits poorer performance than SMOTER and RUS. Previous research has noted the tendency for ROS-based 510 models to overfit, due to the high number of duplicate samples (Yap et al., 2014). RUS, despite using considerable less training data for each individual learner, is not as prone to overfitting as ROS. The RUS-Bagging models consistently outperform the RUS-RWB models; this may be due to the repeated resampling, thus RUS-Bagging uses much more of the original training samples, while RUS-RWB only uses 20% of the original data."

RC2-11. There are few typos and referencing issues. Please revise the manuscript for this. A few examples are provided below:

Line 4: "compare three resampling;" I think it is missing a word! Table 3: second column. Do you mean "variable" or "feature"? Line 113: please include the term "individual learners". Line 163: comma is missing after the reference. Line 164: the reference seems to be in the middle of the sentence. I noticed you do this often. Please try to minimize this to enable smother flow of the idea. Line 168: "for handling imbalance data. . ." do you mean "imbalanced". Line 170: "that featuring each. . .". Please fix. Please italicize all symbols or feature names, such as $N$, $theta$, $PI$, $CE$, etc., across the manuscript. Line 205: the reference format needs a fix. Line 255 to 260: please re-write the ensemble learning summary as discussed in the major comment. Line 262: "Ensembles members are.." please remove the "s". Line 310: do you mean "regional flood quantiles"?.

AC2-11. Thank you for pointing out these typos - we have corrected these issues in the revised version of the manuscript.

References [1] E. Snieder, R. Shakir, and U. T. Khan, "A comprehensive comparison of four input variable selection methods for artificial neural network flow forecasting models," J. Hydrol., vol. 583, p. 124299, Apr. 2020. [2] B. W. Yap, K. A. Rani, H. A. A. Rahman, S. Fong, Z. Khairudin, and N. N. Abdullah, "An Application of Oversampling, Undersampling, Bagging and Boosting in Handling Imbalanced Datasets," Lect. Notes Electr. Eng., vol. 285 LNEE, pp. 13–22, 2014. [3] M. Bach, A. Werner, J. Ążywiec, and W. Pluskiewicz, "The study of under- and over-sampling methods' utility in analysis of highly imbalanced data on osteoporosis," Inf. Sci. (Ny)., vol. 384, pp. 174–190, Apr. 2017. [4] J. F. Díez-Pastor, J. J. Rodríguez, C. I. García-Osorio, and L. I. Kuncheva, "Diversity techniques improve the performance of the best imbalance learning ensembles," Inf. Sci. (Ny)., vol. 325, pp. 98–117, Dec. 2015. [5] D. Opitz and R. Maclin, "Popular Ensemble Methods: An Empirical Study," J. Artif. Intell. Res., vol. 11, pp. 169–198, Aug. 1999. [6] G. Brown, J. Wyatt, R. Harris, and X. Yao, "Diversity creation methods: A survey and categorisation," Inf. Fusion, vol. 6, no. 1, pp. 5–20, 2005. [7] M. Galar, A. Fernandez, E. Barrenechea, H. Bustince, and F. Herrera,

"A review on ensembles for the class imbalance problem: Bagging-, boosting-, and hybrid-based approaches," IEEE Transactions on Systems, Man and Cybernetics Part C: Applications and Reviews, vol. 42, no. 4. pp. 463–484, 2012. [8] J. F. Díez-Pastor, J. J. Rodríguez, C. García-Osorio, and L. I. Kuncheva, "Random Balance: Ensembles of variable priors classifiers for imbalanced data," Knowledge-Based Syst., vol. 85, pp. 96–111, Sep. 2015. [9] N. Moniz, R. Ribeiro, V. Cerqueira, and N. Chawla, "SMOTEBoost for Regression: Improving the Prediction of Extreme Values," in 2018 IEEE 5th International Conference on Data Science and Advanced Analytics (DSAA), 2018, pp. 150–159.

Please also note the supplement to this comment:
https://hess.copernicus.org/preprints/hess-2020-430/hess-2020-430-AC2-supplement.pdf

---

## Author Comment (AC3) · 1 Jan 2021

RC1-1. This manuscript compares different resampling methods and different ensemble- building techniques to improve ANN-based flood forecasts (high stream-flow). Those resampling methods and ensemble techniques are also combined, resulting in a total of 16 variants, that are compared to a base model. The base model is a classic Mul- tilayer Perceptron with 10 hidden neurons and 25 input variables, trained using a stop training approach and the Levenberg-Marquardt algorithm. All the 16 variants and the base model are applied to simulate (I think) the streamflow for two rivers in Canada.

The manuscript is very well written, well organized and very clear. However, I am sorry to say that I find the originality and contribution to be very low, too low in my opinion for a publication in HESS. The research was certainly conducted with great care, but ensemble techniques have already been used for quite a while in hydrology (both in the hydro-informatics community and beyond).

AC1-1. Thank you for the positive comments on our manuscript. However, we would like to clarify the novelty and contribution of the research presented in our manuscript:
+ Extensive comparisons of resampling and ensemble methods (independently and combined) to address the data imbalance problem in data-driven hydrological models. The authors restate the novelty of embedding resampling methods in ensemble methods (as opposed to resampling as preprocessing), which has not previously been studied for hydrological stage forecasting.

+ Many of the combination methods proposed in our work encourage diversity-in-learning, which distinguishes the algorithms from previous work on simple Bagging or boosting methods.

+ Comparison of two watersheds that have different dominant hydrological processes, spatial and temporal scales.

+ The heteroskedastic nature of flow forecasting models, which our work attempts to address, is a persistent issue in the field of hydroinformatics.

+ A particular focus on high stage (rather than the entire timeseries) to assist in early warning systems or flood forecasting.

+ While some previous use in the broader machine learning literature, some methods, adapted and implemented in the manuscript, are a first in hydrology for flood forecasting.

+ To our knowledge, the variations of SMOTER, SMOTER-AdaBoost, and LSBoost with resampling developed in our research are novel implementations.

+ Even though some methods, and resampling or ensemble techniques, have been used in hydrological studies, a systematic comparison, as presented in our manuscript, is still needed to properly evaluate their efficacy, particularly for high flows.

RC1-2. While the authors mention that most of the resampling and ensemble techniques they used are not common in hydrology, this is still just an additional application of existing techniques, some of which have already been compared. Further, there is very little discussion and analysis of the results.

AC1-2. We apologise that the depth of analysis in the original submission was not found to be thorough enough. We have expanded the analysis and discussion in addressing specific comments made by the other reviewers. The revised manuscript will have a longer discussion on the more important points. In summary, these include: + A formalised grid-search of ensemble size for base ensemble methods (with no resampling), which is discussed in greater detail in our response to AC1.13.

+ Additional analysis of the relative effects of the selection of the high stage threshold value (ranging from 50 to 90th percentile) for Bagging and SMOTER-Bagging.

+ Added an evaluation of the relative effects of ensemble size (ranging from 2-100) for Bagging and SMOTER-Bagging.

+ Added additional statements elaborating on the concept of diversity-in-learning and the role and effects of diversity attained using combined resampling - ensemble methods.

RC1-3. The author conclude that boosting methods provide only marginal improvements, they offer very little explanation.

AC1-3. The main finding of this research is that common ensemble methods, when combined with methods with resampling methods that increase the representation of high flow samples in the training distribution, only offer marginal improvements to model performance on high flows. Basic ensemble methods (with no resampling) are included

as a reference point, to quantify the improvements produced by the added resampling. The improvements of ensemble methods over the single model scenario is not discussed in detail, as this is already well established in existing research.

Additional analysis and discussion, as requested by Reviewer 2, has now been added; please refer to Figs 6, 11, and 12, and the associated text in the attached supplement.

RC1-4. In addition, I am not convinced that improving only specifically high flows, sometimes at the expense of what the authors call "typical flows", is the way to go. I also fail to see it as a major contribution to the hydrological sciences, worthy of publication in an international journal.

AC1-4. As stated in text, many studies have specifically noted poor performance of data-driven flow forecasting models on underrepresented flows (i.e., when the dataset is imbalanced, and in practice this may be high flows, as explored in this present research, or low flows, which may suffer from similar imbalance problems, and hence, the same methods may be applied). This issue of poor performance of flow forecasting models on underrepresented flows is the main thrust for the present research - a limitation that has been identified in existing research, and resampling or ensemble techniques (independently, or combined) is one way to address this problem.

The objective of this research is to obtain homoscedastic residuals across different flow values. However our results indicate a notable trade-off between typical and high flow performance. We believe that in certain circumstances such as a high flow warning system, such a trade-off would be desirable; in such a scenario, the discrete values of typical flows are of little importance, so long as they do not result in false positive warnings.

RC1-5. I will still detail some major and minor comments below, but my main concern regarding this paper is the level of the contribution, that I unfortunately find too low.

AC1-5. As mentioned above, we think there is a need for a systematic comparison

of resampling and ensemble methods within hydrology to address the data imbalance problem, particularly for high flows, since they are important for accurate flood forecasting models. While resampling is commonly used in flow forecasting studies, the SMOTER algorithm has only been used in a few instances and never for regression-based hydrological forecasting. Moreover, the resampling in such studies typically takes the form of preprocessing; in our studies, resampling methods are embedded within the ensemble algorithms. Such combinations appear in machine learning literature (mainly in classification studies) but, to our knowledge, ours is the first study to assess the effects of these algorithms on high flow performance and among the first applications in hydroinformatics. Combined resampling - ensemble learning algorithms achieve greater diversity-in-learning than either respective standalone method, hence have the potential to produce better performing models. The specific resampling algorithms chosen increase the influence of high flows in the model training process, thus addressing a common weakness of data-driven flow forecasting models: poor accuracy on high flows.

RC1-6. Detailed comments (apart from the contribution/originality issue): Abstract, lines 1-2: The affirmation "(. . .) are increasingly used for operational flood warning systems" should be supported by references to operational flood forecasting systems (not journal articles but documentation from company web- sites or government websites). At the moment, the affirmation is not only un- supported, but also opposite to my experience. To the best of my knowledge, there are very, very few operational agencies who use ANNs for flood forecasting, despite their use in research for more than 25 years. Also, this affirmation in the abstract somewhat contradicts page 10 lines 157-158 ("Consequently, such models may not be suitable for flood related applications such as flood warning systems").

AC1-6. We apologise for incorrectly inferring high commercial use of data-driven flood forecasting. The text has been modified to emphasise that such models have increasingly been featured in hydrological research, not necessarily used in practice: "Datadriven flow forecasting models, such as Artificial Neural Networks (ANNs), are increasingly featured in research for their potential use for operational flood warning systems."

The statement on page 10 aims to address the common claim that data-driven flow forecasts have high potential for flow forecasting applications; however, often such claims do not explicitly evaluate the performance of such models on high flows. When studies explicitly evaluate high flow performance for these models, they are often found to be lacking, especially relative to the performance on low or typical flows. Hence, there is a need to evaluate the ability of preprocessing and/or ensemble methods to improve model performance for high flows (which are important for flood forecasting and early-warning systems).

RC1-7. Page 2 lines 24-30: One of the causes of errors when simulating high flows in northern locations such as Canada is the occurence of ice jams. Ice jams are very common and not accounted for in any way by typical hydrological models. Maybe it is possible to account for them using ANNs, but I'm not sure. I think this is one major aspect that should have been present in the manuscript, both on page 2 but also when presenting the Bow River and Don River (it would be important that those rivers be ice jams free, otherwise you have to account for that). Another issue regarding high streamflow that is not discussed by the authors is the fact that those readings are extrapolations from the rating curve. The rating curve of a gauging station is typically constructed with very few (if any) observations of very high streamflow. Therefore, this part of the rating curve is very uncertain, and this is what we use to obtain "observations".

AC1-7. We apologise for the lack of clarity in the text about ice jams. Indeed ice jams were unaccounted for in this work: periods during which ice jams would occur in either watershed were removed from the data. Specifically, data from November to April and November to December were removed from the Bow and Don River models, respectively. This has been clarified in text: "Data from November to April and November to December were removed from the Bow and Don, respectively, datasets prior to any

analysis; these periods are associated with ice conditions."

The authors recognise the uncertainty associated with stage-discharge transformations. While we utilise language of 'typical' and 'high flows', only stage observations are used as model input or target features. Predicting stage directly (rather than flow) is sufficient for application flood early warning systems; thus the uncertainty associated with the stage-discharge transformation does not need to be considered. If discharge is required, the subject models could be reconfigured, thus the ANN would implicitly model stage-discharge uncertainty, or discharge could be calculated in post-processing, in which case it would be recommended that uncertainty be estimated and communicated with discharge forecasts. We have also changed the terminology in the manuscript, replacing all occurrences of "flows" with "stage", where appropriate.

RC1-8. Page 6, Table 2 and also Page 7 line 128: Why did you use the Levenberg-Marquardt algorithm? Although it is a popular algorithm, it was shown to have oscillation problems around local minima. I think the use of this algorithm should be better justified. See for instance Kwak et al. (2011)

AC1-8. The LM algorithm was selected because of its popularity, speed of convergence, and reliability [1]. Its suitability was confirmed based on a manual comparison with other available training algorithms in MATLAB for the baseline model. Existing literature has favourably described the LM algorithm, specifically for its ability to escape local minima in the error surface [2, 3]. We have added justification for using the LM algorithm in text as follows: "The Levenberg–Marquardt algorithm was used to train the base models, because of its speed of convergence and reliability [1-3]."

RC1-9. Page 6 line 116: What is the forecasting horizon? You mention the word "predict" here, but from reading the manuscript it seems like you perform simulations. If they are really forecasts, I think the forecasting horizon should be specified.

AC1-9. We apologise for the lack of clarity surrounding the forecast horizon. The forecast horizon is specified in text: "For both rivers, the input variables are used to

forecast the target variable 4 timesteps in advance, i.e., for the Bow River, the model forecasts 24 hours in the future, whereas for the Don River, the model forecasts 4 hours in the future."

Our models use previous timesteps to predict future data (e.g., $Q_{t+4}$ is predicted using $Q_t$, $Q_{t-1}$, $Q_{t-2}$). All models are calibrated using 80% of the available data while 20% of the data is isolated from model calibration and reserved for testing. Using a 10-fold cross-validation scheme, all of the data is included for testing exactly 3 times, across 10 different ensembles. The performance of the 10 ensembles are averaged or visualised in boxplots. In other words, when the models are "predicting" the testing data, they are doing so without "seeing" the test data - historical (i.e., previous) data is used to forecast future data. These predictions are compared to the observations for performance evaluation. Our models can easily be deployed in a real-world scenario as stage, precipitation, and temperature data are all instantaneously available online, and lagged versions of these data are used to forecast the future state of the system.

RC1-10. Page 10 line 170: I think there is a mistake here "(. . .) studies that featuring each (. . .)"

AC1-10. Thank you for identifying this mistake; it has been corrected.

RC1-11. Page 11: the distinction between RUS and ROS seems extremely thin to me.

AC1-11. You are correct - the distinction between RUS and ROS is minor. As stated in-text, RUS undersamples data as to achieve a balanced training set with no duplicates whereas ROS creates includes all available data and creates duplicates. Some studies have compared the two sampling techniques, however neither one of these two methods consistently outperforms the other, thus both are included in our study [4-6]. As requested by Reviewer 2, we will include pseudo-algorithms of each method in the revised manuscript to clarify the difference. The algorithms are attached in the supplement.

[Figure]

RC1-12. Page 13 lines 256-257: The definition of ESP that you provide here corresponds to Extended Streamflow Prediction, as per Day (1985), not Ensemble streamflow predictions. Ensemble streamflow forecasts (or predictions) can be obtained by a variety of manners, including feeding a hydrological model with dynamical meteorological ensemble forecasts.

AC1-12. We apologise for the confusion caused by this statement. This definition has been removed from the text.

RC1-13. Page 14: Why 20 members?

AC1-13. The ensemble size of 20 used in the original manuscript was determined based on an informal trial-and-error search. We deliberately used the same ensemble size for each method for the sake of comparison. In the revised manuscript, we have included a formalised ensemble size grid-search for ensemble sizes ranging from 2-100 for each of the base ensemble methods (i.e., with no added resampling). This analysis reveals that each ensemble method has a different optimum ensemble size; however, we decided to maintain a fixed ensemble size of 20 between all methods. Please refer to Fig. 6 and associated text in the attached supplement.

RC1-14. Results: Why do you aggregate the ensemble into a deterministic simula- tion and therefore evacuate the information about the uncertainty? Why not use ensemble-based performance assessment criterion such as the Continuous Ranked Probability Score, logarithmic score, etc.?

AC1-14. Thank you for this recommendation. The ensembles were combined for two reasons. Firstly, it allows for comparison against the single model scenario. Next, boosting methods are designed to be aggregated; AdaBoost ensemble predictions must be made using the weighted mean of the ensemble predictions and LSBoost uses a weighted sum combiner. Thus, ensemble performance criteria are only valid for uniformly weighted ensembles (RWD and Bagging). Your comment is absolutely correct in that the capability of generating a spread of predictions is an advantage

of these methods. However, for consistency, ensembles were combined into discrete predictions for all cases. That is, ensemble-based metrics are not applicable for all methods used in this research, and the only way for direct comparison is to use a "combined" metric.

RC1-15. Page 19 lines 457-459: How can overtraining happen if you have used stop training? I think this has to be explained more.

AC1-15. Thank you for identifying the point of confusion. Stop-training ensures that a model trained on a 'training' data partition achieves good generalisation on the 'validation' partition. However, this does not guarantee that the generalisation will carry to the independent 'test' partition. For example, if small and similar 'training' and 'validation' subsets are used and a large 'test' set is used, the model could very well be overfitted, despite the use of a stop-training criterion for determining the number of training epochs.

RC1-16. Page 21, Figure 7: From a general perspective, I don't see how decreasing the quality of simulation for typical flow values could be positive, even if the simulation of high flows is improved. Typical and especially very low flows are also important.

AC1-16. We believe that our statement is true for very specific applications, such as data-driven early flood warning systems (where the primary interest is on high flows that may lead to floods, rather than on typical or low flows). Moreover, the objective of this research was to reduce forecast error on high flows, not necessarily at the expense of typical flow performance; the trade-off between high and typical flow performance is simply the observed effect of the methods under study and not the objective. Finally, it is important to note that the methods for improving high flows examined in this research are transferable to other rare observations, such as low flows.

RC1-17. Page 26 lines 531-536: The analysis and discussion are very thin.

AC1-17. Thank you for this feedback. We have added a citation [7] to support the

tendency for boosted models to overfit. The brief discussion here is simply intended to allude that the improvements produced by boosting in this study being relatively lower than what is observed in other studies could be owed to the use of ANNs as the base learner. The vast majority of studies that use boosting utilise decision trees as the base learner, thus the outcome of these studies may not provide a reliable comparison. A formal comparison between different base learners for ensemble methods is not a goal of this study.

RC1-18. Page 26, section 4.2: I disagree with the idea of fine tuning the hyper parameters differently and specifically for each model, as it would violate the ceteris paribus principle, making it difficult to isolate and compare the influence of ensemble techniques and resampling techniques.

AC1-18. We did not intend to violate the ceteris paribus principle with the referenced statement. We made an effort to keep overlapping hyperparameters equal between different ensemble models. However, for the same base learner (a simple ANN) a standalone model and boosted model will have different complexities, hence different effective degrees of freedom (DOF). So, the question becomes should the individual model topology be kept equal (e.g., number of hidden neurons) or should the DOF be made equal? Moreover, some hyperparameters (e.g., Learning Rate in LSBoost) are specific to the method and have no counterpart among the other methods. Ideally, the comparison in this paper would be carried out for varying hyperparameter values and at varying degrees of model complexity; however, such a comparison would have a large computational cost and was considered beyond the established scope.

RC1-19. Page 26-27 lines 562-563: Ensembles are already quite common in ANN-based flow forecasting model, so this is not a very useful recommandation.

AC1-19. Thank you for this feedback. We acknowledge that ensembles are widely used in flowcasting, however we believe that they are worthy of investigation because of (1) the amount and variety of different ensemble methods and (2) the potential for

their combination with resampling methods. The outcome of the comparison presented in this work reveals that the combinations of ensemble and resampling methods under study do not outperform simple, proven ensemble methods with no resampling. In some cases, resampling can be used to produce marginal improvements in high flow performance, at a disproportionate trade-off with typical or low flow performance. Thus, our recommendation of using ensemble methods does not suggest that their superiority over single models is a novel finding; rather, that the combination of resampling and ensemble methods, which are widely used in machine learning literature, do not result in meaningful improvements when compared to the same ensemble methods with no resampling methods.

Additionally, we believe that our recommendation to the employ of simple ensemble methods such as Bagging are warranted, as despite their benefits being well established in this field of research, their use should be considered a minimum requirement for data-driven hydrological modelling.

References

[1] N. Lauzon, F. Anctil, and C. W. Baxter, "Clustering of heterogeneous precipitation fields for the assessment and possible improvement of lumped neural network models for streamflow forecasts," Hydrol. Earth Syst. Sci., vol. 10, no. 4, pp. 485–494, Jul. 2006.

[2] H. R. Maier and G. C. Dandy, "Neural networks for the prediction and forecasting of water resources variables: A review of modelling issues and applications," Environ. Model. Softw., vol. 15, no. 1, pp. 101–124, Apr. 2000.

[3] H. Tongal and M. J. Booij, "Simulation and forecasting of streamflows using machine learning models coupled with base flow separation," J. Hydrol., vol. 564, pp. 266–282, Sep. 2018.

[4] B. W. Yap, K. A. Rani, H. A. A. Rahman, S. Fong, Z. Khairudin, and N. N. Abdullah,
"An Application of Oversampling, Undersampling, Bagging and Boosting in Handling Imbalanced Datasets," Lect. Notes Electr. Eng., vol. 285 LNEE, pp. 13–22, 2014.

[5] M. Bach, A. Werner, J. Åżywiec, and W. Pluskiewicz, "The study of under- and over-sampling methods' utility in analysis of highly imbalanced data on osteoporosis," Inf. Sci. (Ny)., vol. 384, pp. 174–190, Apr. 2017.

[6] J. F. Díez-Pastor, J. J. Rodríguez, C. I. García-Osorio, and L. I. Kuncheva, "Diversity techniques improve the performance of the best imbalance learning ensembles," Inf. Sci. (Ny)., vol. 325, pp. 98–117, Dec. 2015.

[7] A. Vezhnevets and O. Barinova, "Avoiding Boosting Overfitting by Removing Confusing Samples," in Machine Learning: ECML 2007, vol. 4701 LNAI, Berlin, Heidelberg: Springer Berlin Heidelberg, 2007, pp. 430–441.

Please also note the supplement to this comment:
https://hess.copernicus.org/preprints/hess-2020-430/hess-2020-430-AC3-supplement.pdf

[Figure]

**Supplement:**

[Figure]

Figure 6. Test MSE across ensemble size for RWB (red), Bagging (blue), AdaBoost (yellow), and LSBoost (green) for the Don (left) and Bow River (right).

"Fig. 6 illustrates the change in test performance as the ensemble size increases from 2 to 100 for each river. This grid search is performed only for the base ensemble methods (RWB, Bagging, AdaBoost, and LSBoost) without any resampling. The Bow River plot indicates that AdaBoost and LSBoost tend to favour a small ensemble size (2-15 members), whereas the generalisation of RWB and Bagging improves with a larger size (>20 members). The performance of LSBoost rapidly deteriorates as the ensemble size grows, likely as the effects of overfitting become more pronounced. Similar results are obtained for the Don, except that RWB, Bagging, and AdaBoost all improve with larger ensemble size, while LSBoost does not offer competitive performance for any ensemble size. Again, a larger ensemble size (>20 members) produces favourable MSE."

[Figure]

Figure 11. Calibration and test MSE ratio between Bagging and SMOTER-Bagging models for the Bow (a) and (c) and Don (b) and (d) Rivers across high stage threshold values ranging from 50% to 90%.

"As discussed in Sect. 2.1, a fixed threshold is used to distinguish between high and typical stages. Fig. 11 shows the effects of the fixed threshold increasing from the 50th to 90th percentile of the stage distribution. These plots show the relative effects of SMOTER-Bagging compared to simple Bagging. A performance ratio greater than 1 indicates that the SMOTER-Bagging model has greater error compared to the Bagging model, 1 indicates that they have the same performance, and less than 1, improved performance. The error (MSE) is presented for all stages as well as the TS and HS subsets. The calibration plots illustrate an asymmetric trade-off between HS and TS error. For a given $\theta_{HS}$ value, the error ratio of the TS subset increases more than the decline in HS error. More importantly, the improvements in HS performance obtained in calibration are considerably less pronounced in the test dataset, despite a loss in TS performance".

[Figure]

Figure 12. Test MSE ratio between Bagging and SMOTER-Bagging models for the Bow (a) and the Don (b) across ensemble size.

"Fig. 12 illustrates the effects of varying the ensemble size, thus, number of resampling repetitions, for the SMOTER-Bagging model, relative to the simple Bagging model. The plot shows the relative improvement in HS produced by the SMOTER resampling as the ensemble size increases, reaching a steady value at an ensemble size of approximately 70 for both models. This is larger than that required for the simple Bagging model to reach steady performance, indicating that SMOTER requires more resampling than simple resampling with replacement in order to reach stable performance. Consistent observations made from Fig. 11, an asymmetric trade-off between typical and high stage performance is noted."

**Appendix B: Pseudocode**
* * *
**Algorithm 1** Random undersampling
* * *
**Require:**

    Set S containing X input features and Y observations, $(x_1, y_1), ..., (x_m, y_m)$

    High stage threshold, $\theta$

    $S_{TS} = S$ where $Y < \phi_{TS}$

    $S_{HS} = S$ where $Y \geq \phi_{HS}$

    $S'_{TS} \leftarrow sample(S_{TS}, N_{HS})$

    $S'_{HS} \leftarrow sample(S_{HS}, N_{HS})$

    $S' = S'_{TS} \bigcup S'_{HS}$
* * ** * *
**Algorithm 2** Random oversampling
* * *
**Require:**

    Set S containing X input features and Y observations, $(x_1, y_1), ..., (x_m, y_m)$

    High stage threshold, $\theta$

    $S_{TS} = S$ where $Y < \phi_{TS}$

    $S_{HS} = S$ where $Y \geq \phi_{HS}$

    $S'_{TS} \leftarrow sample(S_{TS}, N_{TS})$

    $S'_{HS} \leftarrow sample(S_{HS}, N_{TS})$

    $S' = S'_{TS} \bigcup S'_{HS}$